# Effect of the Revisit Interval and Temporal Upscaling Methods on the Accuracy of Remotely-Sensed Evapotranspiration Estimates

Joseph G. Alfieri[1], Martha C. Anderson[1], William P. Kustas[1], Carmelo Cammalleri[2]

5    [1]US Department of Agriculture, Agricultural Research Service,
      Hydrology and Remote Sensing Laboratory, Beltsville, MD, USA
    [2]Joint Research Centre, European Commission, Ispra, Italy

*Correspondence to*: Joseph G. Alfieri (joe.alfieri@ars.usda.gov)

**Abstract.** Accurate spatially distributed estimates of actual evapotranspiration (ET) derived from remotely sensed data are critical to a broad range of practical and operational applications. However, due to lengthy return intervals and cloud cover, data acquisition is not continuous over time, particularly for satellite sensors operating at medium (~100 m) or finer resolutions. To fill the data gaps between clear-sky data acquisitions, interpolation methods that take advantage of the relationship between ET and other environmental properties that can be continuously monitored are often used. This study sought to evaluate the accuracy of this approach, which is commonly referred to as temporal upscaling, as a function of satellite revisit interval. Using data collected at 20 Ameriflux sites distributed throughout the contiguous United States and representing 4 distinct land cover types (cropland, grassland, forest, and open canopy) as a proxy for perfect retrievals on satellite overpass dates, this study assesses daily ET estimates derived using 5 different reference quantities (incident solar radiation, net radiation, available energy, reference ET, and equilibrium latent heat flux) and 3 different interpolation methods (linear, cubic spline, and hermite spline). Not only did the analyses find that the temporal autocorrelation, i.e. persistence, of all of the reference quantities was short, it also found that those land cover types with the greatest ET exhibited the least persistence. This carries over to the error associated with both the various scaled quantities and flux estimates. In terms of both the root mean square error (RMSE) and mean absolute error (MAE), the errors increased rapidly with increasing return interval following a logarithmic relationship. Again, those land cover types with the greatest ET showed the largest errors. Moreover, using a threshold of 20% relative error, this study indicates that a return interval of no more than 5 days is necessary for accurate daily ET estimates. It also found that the spline interpolation methods performed erratically for long return intervals and should be avoided.

## 1 Introduction

As one component of a complex network of interconnected processes, evapotranspiration (ET) is influenced by numerous factors such as available energy, soil moisture, vegetation density, and humidity (Farquhar and Sharkey 1982; van de Griend and Owe 1994; Alves and Pereira 2003; Alfieri et al. 2007). For example, the amount energy available to drive ET depends on atmospheric properties, such as humidity and aerosol content, which influence atmospheric transmissivity (Brutseart 1975; Bird and Riordan 1986). The available energy is also controlled by surface properties, such as the type and density of vegetation cover and soil moisture, which influence not only the surface albedo and emissivity (Wittich 1997; Asner etal. 1998; Myneni et al.1989; Song et al. 1999; Lobell and Asner 2002), but also impact the amount of energy conducted into the ground (Friedl and Davis 1994; Kustas et al. 2000; Abu-Hamadeh 2003; Santanell and Friedl 2003). Moreover, the magnitude of the moisture flux can vary over a range of timescales in response to changes in the environmental conditions influencing ET. One example of this, which has been pointed out by Williams et al. (1998), Scott et al. (2014), and others, is the rapid and often persistent change in ET in response to a rain event.

Because it is a fundamental linkage between numerous biogephysical and biogeochemical processes, accurate information regarding evapotranspiration (ET) is critical for a broad range of scientific and practical applications with significant social, economic, and environmental impacts. For example, reliable information about ET is essential for accurately forecasting weather and assessing the impacts of changing climate (Katul et al., 2012; Wang and Dickinson, 2012); monitoring and mitigating the adverse effects of extreme weather events such as drought (Anderson et al., 2007, 2011, 2016; Otkin et al. 2016); and, identifying and predicting the changes in both the biogeographical characteristics of ecosystems  and the services they provide in response to changing environmental conditions (Hawkins and Porter, 2003; Kreft and Jetz, 2007; Midgley et al. 2002). However, as pointed out by Seguin and Itier (1983), Abdelghabi et al. (2008), and Anderson et al. (2012), among others, perhaps the most important application of ET data is providing information critical to satisfying the competing demands for scare water resources.

Already, the competing demands for fresh water by agricultural, industrial, and urban consumers exceed the available supply for nearly one-third of the world's population (Qadir et al. 2003) and it is predicted that number will increase to more than two-thirds of the global population in the coming decades (Wallace, 2008; Vörösmarty et al., 2010). To meet the current and future demand for water, resource managers and other policymakers must make informed decisions regarding the needs of competing stakeholders when allocating limited water resources in order to maximize their effective use. In the case of irrigated agriculture, which is the largest consumer of fresh water and accounts for 1200 km$^3$ or approximately 85% of annual current water use (Drooger et al., 2010; Thenkabail et al., 2010), the need for water is largely driven by evaporative loss. Thus, ET measurements are needed not only to monitor evaporative water loss and determine crop irrigation needs, it is also needed to develop the irrigation techniques and management practices necessary to ensure the efficient use of water in agricultural environments (Howell, 2001; Schultz and Wrachien, 2002; Gordon et al., 2010; de Fraiture and Wichelns, 2010).

While in situ observations are invaluable for some of these applications, many of them require spatially distributed measures of ET at field to continental scales that cannot be supplied by the existing flux measurement infrastructure. Remote sensing-based approaches are the only viable mean for monitoring ET over this continuum of scales (McCabe et al., 2008; Kalma et al., 2008; Gonzalez-Dugo et al., 2009). As discussed by Anderson et al. (2012), any comprehensive program for monitoring water resources will by necessity use remote sensing data collected by multiple platforms at a range of spatial and temporal scales.

Nonetheless, remote sensing is not without limitations. Chief among these is the infrequent acquisition of the medium to high-resolution imagery needed as input for remote sensing-based models to determine ET. This infrequent acquisition of imagery is due to both lengthy return intervals and the presence of cloud cover (Ryu et al., 2012; van Niel et al., 2012; Cammalleri et al., 2013). To provide temporally continuous ET estimates, the moisture flux during the period between data acquisitions is often estimated using an interpolation technique commonly referred to as temporal upscaling. This well-established approach, which can be applied at either sub-daily or daily time steps, estimates the moisture flux as the product of some reference quantity ($\chi$) and its associate scaled metric ($f$) according to:

$$\widehat{\mathrm{ET}}_t = \chi_t f_t \tag{1}$$

where $\widehat{\mathrm{ET}}$ is the estimated ET and $t$ indicates the time period of the estimate. While it is typically related to the moisture flux, $\chi$ is a quantity that can be measured or estimated more readily than the moisture flux itself. The scaled metric is the ratio between $\chi$ and the moisture flux. For example, it is quite common to estimate ET expressed in terms of the latent heat flux ($\lambda E$) using the available energy ($A$), here defined as the net radiation less the soil heat flux, as the reference quantity and evaporative fraction ($f_A$) as the scaled metric (*e.g.* Crago and Brutsaert , 1996; Bastiaanssen et al., 1998; Suleiman and Crago, 2004; Colaizzi et al., 2006; Hoedjes et al. 2008; van Niel et al., 2011; Delogu et al., 2012).

For the periods between data retrievals, $f$ is estimated via interpolation. As a result, this approach is predicated on the assumption that $f$ is self-preserving, *i.e.* it is constant or nearly constant, and thus varies only slowly over time (Brutsaert and Sugita, 1992; Nichols and Cuenca, 1993; Crago, 1996). In order to conform to this assumption, the components of the radiation or energy budget are often selected as $\chi$ such that $f$ is an analogue of evaporative fraction. Examples of these quantities include the incident solar radiation ($K\downarrow$; Jackson et al. 1983; Zhang and Lemeur, 1995), or extraterrestrial solar radiation ($K\downarrow_{\mathrm{TOA}}$; Ryu et al. 2012). However, a number of recent studies indicate the assumption of self-preservation is only approximate for these quantities. For example, both Gentine et al. (2007) and Hoedjes et al. (2008) showed that the self-preservation of evaporative faction is sensitive to soil moisture conditions and fractional vegetation cover. Similarly, Lhomme and Elguero (1999) and later Van Niel et al. (2012) showed that the degree of self-preservation can be influenced by cloud cover. As such, the assumption of clear-sky conditions is a significant potential source of error in the ET estimates that must be considered when utilizing or evaluating temporally upscaled moisture flux data (van Niel et al., 2012; Peng et al., 2013; Cammalleri et al., 2014). Other studies have focused on using a quantity derived from the local meteorological conditions as $\chi$ because it would consider many of the meteorological factors that influence the moisture flux. For example, Tasumi et al. (2005) proposed using the reference ET for alfalfa ($\mathrm{ET}_r$) as $\chi$; later, Allen et al. (2007) proposed using the

standardized reference evapotranspiration ($ET_0$) as $\chi$. In both cases, the resulting $f$ is equivalent to a crop coefficient and would share its characteristics. As a result, $f$ derived from $ET_r$ or $ET_0$ can be treated in much the same fashion as a crop coefficient and assumed to be nearly constant changing only slowly with time (Colaizzi et al., 2006; Chavez et al., 2009).

By assessing the error introduced into ET estimates by temporal upscaling under realistic conditions, this study sought to achieve two goals: i. to provide insights into the relative strengths of the differing temporal upscaling approaches, and *ii.* to determine the maximum return interval threshold for obtaining acceptable estimates of daily ET. Specifically, this study uses in situ measurements collected over a variety of land cover types as a proxy for remotely-sensed data to evaluate the impact of multiple reference quantities, interpolation techniques, and revisit intervals on the estimated daily moisture flux. The study focuses on daytime mean data to evaluate temporal upscaling at a daily time step. It also assumes perfect retrieval of the flux; in other terms, no error was introduced into ET data to approximate the error or uncertainty in the estimates of ET from the remote sensing-based models. Since any errors in the remote sensing-based ET estimates propagate into the calculation of $f$ and the subsequent temporal upscaling, this analysis represents the best-case scenario. The following section provides an overview of the field measurements along with the reference quantities, interpolation techniques, and evaluation methods used in this study. Section 3 provides a discussion of the results of this study while the final section encapsulates the conclusions that can be drawn from those results.

## 2. Methods

### 2.1. Datasets

Data, including local meteorological conditions, surface fluxes, and surface conditions (Table 1), collected at numerous sites within the Ameriflux network (Baldocchi et al., 2001) were used for this study. Specifically, the data were collected at 20 Ameriflux sites (Fig. 1 and 2; Table 2) distributed across the contiguous United States and representing four distinct land cover types. These are *i.* croplands (maize (Zea mays)/soy (Glycine max) rotation); *ii.* grasslands; *iii.* forests (evergreen needleleaf and broadleaf deciduous); and, *iv.* open-canopy (shrubland and woody savanna). Measurements were collected for a minimum of five years at each of the sites selected. Further information regarding the field sites, measurement procedures, and post-processing protocols for Ameriflux is presented in Baldocchi et al., 2001; the data are archived at the Oak Ridge National laboratory and available at http://ameriflux.ornl.gov/.

After forcing closure of the energy balance while maintaining a constant Bowen ratio (Twine et al. 2000) in order to more closely match the characteristics of the output from the models, the 30-minute measurements were used to calculate the various $\chi$ and $f$. Finally, the daytime mean of the fluxes and other necessary quantities were calculated for use in the subsequent analyses. Although it can be taken as nominally as the period between 0800 and 1800 LST, daytime is defined herein as the period between the first and last measurements during a given day when the incident solar radiation exceeded 100 W m$^{-2}$.

## 2.2. Reference Quantities and Scaled Metrics

The first of the $\chi$ derived from meteorological data, $\lambda E_0$, is derived from $ET_0$ which is described by Allen et al. (1998) as the hypothetical ET (or $\lambda E$) from a well-watered grass surface with an assumed height of 0.12 m and albedo of 0.23. It is calculated using a simplified form of the Penman-Monteith equation. For this study, the updated relationship given by Walter et al. (2005) was used:

$$ET_0 = \frac{0.408\Delta(R_n - G) + \gamma\frac{C_n}{(T+273)}U(e_s - e_a)}{\Delta + \gamma(1 + UC_d)} \qquad (2)$$

where $\Delta$ is the slope of the saturation vapor pressure-temperature curve (kPa K$^{-1}$), $R_n$ is the net radiation (W m$^{-2}$), G is the soil heat flux (W m$^{-2}$), $\gamma$ is the psychrometric constant (kPa K$^{-1}$), $C_n$ is a constant (37 °C s$^2$ m$^{-2}$), $T$ is the air temperature (°C), $U$ is the wind speed (m s$^{-1}$), $e_s$ is the saturation water vapor pressure (kPa), $e_a$ is the actual water vapor pressure (kPa), and $C_d$ is a constant (0.24 s m$^{-1}$). This relationship is nearly identical to the one given in Allen et al. (1998); the two formulae differ only with regard to the assumed surface resistance. While the surface resistance is assumed to be 70 s m$^{-1}$ by Allen et al. (1998), it is assumed to be 50 s m$^{-1}$ in the later work. While modest, this modification yields improved results when the daytime moisture flux is calculated on an hourly basis (Walter et al. 2005). The result is converted to $\lambda E_0$ by multiplying by the product of the density of water and the latent heat of vaporization. Similarly, $\lambda E_{eq}$, which can be thought of as the energy-driven moisture flux that is independent of surface resistance, can be expressed according to:

$$\lambda E_{eq} = A\frac{\Delta}{\Delta + \gamma} \qquad (3)$$

with the variables defined as above (McNaughton, 1976; Raupach, 2001).

## 2.3. Interpolation Techniques

In addition to piecewise linear interpolation, two piecewise spline interpolation methods were evaluated as a part of this study, namely cubic and hermite spline interpolation. (These are indicated when necessary hereafter by the subscripts L, S, H, respectively.) In contrast to linear interpolation, which tends to yield accurate results only when the underlying data vary smoothly over time, the splining methods are less prone to error when the observed data change abruptly (Trefethen 2013). Similarly, the more computational complex hermite spline method typically yields more accurate results when the gaps between observed data points are large (DeBoor, 1994).

As the name implies, the piecewise linear interpolation estimates $f$ using a family of $n - 1$ linear relationships defined such that the linearly-interpolated $f$ ($\hat{f}_L$) at time $t$ is determined according to:

$$\hat{f}_{L_i}(t) = f_i + (t_{i+1} - t_i)m_i h \quad t_i \le t \le t_{i+1} \qquad (4)$$

where $n$ is the number of observed data points, $f_i$ is the known $f$ at time $t_i$, $m_i$ is the slope of straight line relationship for the period between $t_i$ and $t_{i+1}$ defined as $m_i = (f_{i+1} - f_i)/(t_{i+1} - t_i)$, and $h$ is the time normalized between 0 and 1 and is

defined as $h = (t - t_i)/(t_{i+1} - t_i)$. The piecewise cubic spline interpolation function is family of $n$ - 1 cubic polynomials defined such that the interpolated $f(\hat{f}_S)$ at time $t$ is determined according to:

$$\hat{f}_{S_i}(t) = f_i + a_i[(t_{i+1} - t_i)h]^3 + b_i[(t_{i+1} - t_i)h]^2 + c_i[(t_{i+1} - t_i)h] \quad t_i \leq t \leq t_{i+1} \tag{5}$$

where the coefficients $a_i$, $b_i$, and $c_i$ are determined by simultaneously solving the series of $n - 1$ equations with the constraints that the interpolation function, as well as its first and second derivatives, must be continuous and pass exactly through the known values of $f$. Similarly, the final interpolation technique, piecewise hermite cubic spline, defines the

$$\hat{f}_{H_i}(t) = (2h^3 - 3h^2 + 1)f_i + (-2h^3 - 3h^2)f_{i+1} + \cdots$$
$$h(h^2 - 2h + 1)(t_{i+1} - t_i)s_i + h(h^2 - h)(t_{i+1} - t_i)s_{i+1} \quad t_i \leq t \leq t_{i+1} \tag{6}$$

where $s_i$ is the slope of the curve at time $t_i$ (De Boor, 1994). For this study, it is calculated according to:

$$s_i = \frac{1}{2}\left(\frac{f_{i+1} - f_i}{t_{i+1} - t_i} + \frac{f_i - f_{i-1}}{t_i - t_{i-1}}\right) \tag{7}$$

and the variables are defined as above (Moler, 2004).

For this analysis, temporal upscaling was conducted at each of the Ameriflux sites using all possible combinations of $f$ and interpolation methods. Specifically, it was conducted with data representing return intervals of up to 32 day generated from the daytime mean data at each site. In order to maximize the robustness of the statistical analysis, all possible realizations - the unique yet equivalent time series that can be generated from the data collected at a particular site while maintaining constant return interval – were considered in the analysis. The total number of possible realizations for a given return interval is equal to the length of the return interval. The individual realizations were generated by beginning the time series on consecutive days.

Again, to emulate the temporal upscaling of flux data derived from remotely sensed-data as closely as possible, efforts were made to ensure that the observations used for the interpolation were collected on clear-sky days. Clear-sky days were identified as those where the daytime mean of the measured $K\downarrow$ was within 25% of the predicted value from a simple radiation model; this threshold was selected based on a preliminary analyses comparing the model results with observations on known clear-sky days. The incident solar radiation was estimated as the product of $K\downarrow_{TOA}$ calculated following Meeus et al. (1991) and atmospheric transmissivity calculated according to Brutsaert (1975). In order to ensure a constant return interval for a given interpolation, if a day was judged to be cloudy, both the observed flux on that day and the estimated flux for those subsequent days derived from it were omitted from the statistical analysis. Although the number of days flagged due to cloudy conditions and omitted from subsequent analyses varied depending on the site and the return interval being modelled, at least 1200 days were considered for each of the analyses at each site.

## 2.4. Statistical Metrics

As discussed by Wilks (2006), persistence, *i.e.* the degree of self-preservation, can be assessed via auto-correlation ($\rho$). For a given lag ($h$), *i.e.* the offset between measurements pairs, the auto-correlation is defined according to:

$$\rho = \frac{\sum_{i=1}^{n-h}[(x_i - \bar{x}_-)(x_{i+h} - \bar{x}_+)]}{\sqrt{\sum_{i=1}^{n-h}(x_i - \bar{x}_-)^2 \sum_{i=1}^{n-h}(x_{i+h} - \bar{x}_+)^2}} \tag{8}$$

where $n$ is the number of data points, $\bar{x}_-$ is the mean of the first $n - h$ data points and $\bar{x}_+$ is the mean of the final $n - h$ data points.

A pair of statistics are used to evaluate the accuracy of the temporal upscaling. The first of these is the root mean square error (RMSE):

$$\text{RMSE} = \sqrt{\frac{1}{n}\sum_{i=1}^{n}(x_i - \hat{x}_i)^2} \tag{9}$$

where $n$ is the number of data points, x is the observed flux, and $\hat{x}$ is the flux predicted by temporal upscaling. However, because the squared difference term in RMSE tends to overemphasize the effects of large errors (Legates and McCabe, 1999; Willmott and Matsuura, 2005; Willmott et al., 2012), the mean absolute error (MAE) was also calculated as follows:

$$\text{MAE} = \frac{1}{n}\sum_{i=1}^{n}|x_i - \hat{x}_i| \tag{10}$$

with the variables defined as above.

Once calculated for the individual the sites, the statistics were aggregated to represent the typical results for a given land cover type. The aggregation was accomplished by calculating the arithmetic means after conducting any necessary transform. For example, both the auto-correlation and RMSE are non-additive quantities that cannot be averaged directly; instead, they must first be transformed into an additive quantity. In the case of the former, the auto-correlation was aggregated by averaging the results for the individual analysis periods at each of the sites after applying a Fisher z-transformation (Burt and Barber, 1996). Similarly, the RMSE data was averaged after first transforming it to the mean square error.

## 3. Results and Discussion

### 3.1 Persistence of Scaled Quantities

Due to its importance in determining the accuracy of the estimates, the persistence or degree of self-preservation exhibited by the various $f$ used in this study was evaluated by determining its autocorrelation function. For each site, the autocorrelation was calculated for each contiguous segment of daytime mean data that was at least 48 days in length (1.5 times the maximum return interval considered herein).

As can be seen in Fig. 3, which shows the mean auto-correlation function for each $f$ and land cover type, all $f$ performed similarly. In all cases, $\rho$ decreased in inverse proportion to $h$, dropping to less than 0.50 within three to ten days. Also, for any given land cover, the mean auto-correlation functions for the analogues of evaporative fraction, namely $f_{K\downarrow}$, $f_{Rn}$, and $f_A$, were statistically indistinguishable from one another based on t-tests conducted at the 95% confidence level. Similarly, no statistically significant difference between the mean auto-correlation functions of $f_{\lambda E0}$ and $f_{\lambda Eeq}$ was found. Nonetheless, there were statistically significant, albeit modest, differences between the auto-correlation functions associated

with $f$ derived from evaporative fraction analogues and those derived from meteorological data. Regardless of land cover, $\rho$ associated with $f_{K\downarrow}$, $f_{Rn}$, and $f_A$, tended to be greater than $\rho$ associated with either $f_{\lambda E0}$ or $f_{\lambda Eeq}$. On average, the difference was approximately 0.03.

The results of this analysis, which are consistent with results of other studies (Farah et al. 2004; Lu et al., 2013) that found significant day-to-day and seasonal variations in evaporative fraction, indicates the long-term persistence of $f$ is very limited. This result also suggests that interpolated values of $f$ may not accurately reflect the actual values and, as a result, may be a key source of error when using temporal upscaling to estimate the moisture flux between image retrievals.

The figure also shows there was significant variability from site-to-site within a given land cover type, particularly for longer lags. Although the specific causes of these differences are not fully understood, there are number of factors that likely contribute. For example, there are difference in both species composition and climate at the various sites. Consider, as an example, the forest class which includes both coniferous and broadleaf deciduous forest. Moreover, the species composition varies even among sites of the same forest type; for example dominant species at the Niwot ridge site are Subalpine fir (Abies lasiocarpa) and Engelmann spruce (Picea engelmannii) while, as the name implies, the dominant species at the Loblolly Pine is loblolly pine (Pinus taeda). At the same time, the mean annual temperature at the forested sites ranged from 1.5 °C to 14.4 °C while the mean annual precipitation varied from 800 mm to 1372 mm. Similarly the mean annual temperature and precipitation at the cropland sites, which are all planted on a rotation of maize and soy, range between 6.4 °C and 11.0 °C and 789 mm and 991 mm, respectively.

Further analysis shows differences in the mean autocorrelation functions exist between land cover types. Regardless of the scaled quantity considered, the mean autocorrelation function decreases most rapidly over forested sites and the most slowly over the open canopy sites. Indeed, if the lag where the mean autocorrelation function reaches some threshold value, e.g. 0.50, is plotted as a function of the mean daytime latent heat flux (Fig. 4), it can be seen that persistence decreases exponentially with the increasing moisture flux. While the underlying cause of this relationship is unclear, it suggests the return interval necessary to achieve accurate estimates of ET via temporal upscaling will be longer over relatively dry regions with a low moisture flux than over regions where ET is high.

**3.2 Accuracy of the Interpolated Scaled Quantities**

Both RMSE and MAE of the interpolated estimates of each $f$ were calculated for all land cover types and return intervals up to 32 days. As can be seen in Fig. 5 and Fig. 6, both metrics behaved similarly; regardless of the land cover type, scaled quantity, or interpolation method considered, the error increased rapidly with increasing return interval until a plateau was reached. In all cases, the RMSE, which increased according to a logarithmic function of return interval, reached 75% of its peak value within five days. For comparison, the mean maximum RMSE for each land cover type was 0.26, 0.28, and 0.17 for croplands, grasslands, forest, and open canopies, respectively. Although it also increased logarithmically, the amount of time needed for MAE to reach 75% of the peak value was more variable, ranging between 5 to 10 days. Again, for purposes of comparison, the mean maximum MAE was 0.22, 0.14, 0.16, and 0.10, respectively, for croplands, grasslands,

forest, and open canopies. Further, MAE increased most rapidly for those land cover types that exhibited the highest moisture flux. The largest error, whether measured in terms of RMSE or MAE, also tended to be associated with the forest and cropland sites where the mean ET was largest.

The results also show that all of the interpolation methods yielded similar results for short return intervals of less than eight days. In contrast, for longer return intervals, both RMSE and MAE of the estimates using the spline interpolation methods were greater than when linear interpolation is used (Fig.5 and 6). Moreover, the error of the estimates tended to much noisier for the spline techniques, particularly the cubic spline method which exhibited periods of very large errors. These large noisy errors, which are most evident in RMSE – perhaps because it is more sensitive to outliers than MAE – are indicative of "overshoot" errors by the spline interpolation. The large errors are also most pronounced for those land cover types that also demonstrated the highest average ET and the lowest autocorrelation

## 3.3 Accuracy of the Latent Heat Flux Estimates

Not unexpectedly, the accuracy of the moisture fluxes estimated via temporal upscaling closely mirrors the accuracy of the interpolated $f$. As was the case with $f$, both the RMSE and MAE of the flux estimates increase rapidly with return interval to a maximum value following a logarithmic function (Fig. 7 and Fig. 8). In the case of RMSE, the maximum error ranged between 31 W m$^{-2}$ and 66 W m$^{-2}$. In the case of MAE, it ranged between 22 W m$^{-2}$ and 54 W m$^{-2}$. Again, the greatest error is associated with the land cover with the highest ET, i.e. forest and cropland.

These plots, like those for $f$, show little difference among the interpolation techniques when the return interval is short. For return intervals longer than about 8 days, however, the spline interpolation techniques, and especially the cubic spline method, can introduce large errors into the flux estimates due to the "overshoot" errors in the interpolation of $f$. These large noisy errors are most evident in the RMSE of forested sites (Fig. 8), but may also be seen to a lesser extent at the cropland sites. Overall, this suggests there is no substantive advantage of using the more computational complex spline techniques over linear regression; rather, the propensity of spline methods to introduce large errors due to interpolation "overshoot" indicates these techniques should be avoided.

The accuracy, and thus utility, of the various $f$ was evaluated while focusing specifically on the results when linear interpolation was used. Regardless of $f$, an intercomparison of the estimated fluxes using t-tests conducted at the 95% confidence level indicated there was no statistically significant difference in either the flux estimates or the error due to temporal upscaling when the return interval was less than eight days. For longer return intervals, analyses using RMSE (Fig. 9) and MAE (not shown), which yielded similar results, indicated the error due temporal upscaling was very similar when $f_{Rn}$, $f_A$, or $f_{\lambda Eeq}$ was used. Indeed, the error introduced using any of these three quantities was statistically identical based on t-tests conducted at the 95% confidence level. Moreover, with the exception of the forest sites, where the error due to temporal upscaling using $f_{K\downarrow}$ was the same as the error introduced by using $f_{Rn}$, $f_A$, or $f_{\lambda Eeq}$, temporal upscaling using $f_{Rn}$, $f_A$, and $f_{\lambda Eeq}$ consistently introduced the least error. For a 10-day return interval, as an example, the percent error introduced by these quantities ranges between 21% and 23% depending on land cover. In contrast, temporal upscaling using $f_{\lambda E0}$ introduced the

greatest error. Again, for a 10-day return interval, the percent error associated with $f_{\lambda E0}$ ranges between 24% and 30% depending on land cover.

The analysis of $D$ reinforces the earlier results. Initially, $D$ decreases rapidly with increasing return interval to less than 0.75 within 3 days and less than 0.50 within 4 to 7 days. This sharp decline in D, which is consistent for all land cover types and $f$ (Fig. 10), indicates there is only moderate agreement between the observed flux and that estimated via temporal upscaling for all but the shortest return intervals. Also, while the moisture flux estimated using $f_{Rn}$ tends to maintain the highest degree of agreement, followed closely by estimates using $f_A$, and $f_{\lambda Eeq}$, the variability in $D$ tends to be modest; for any given land cover type and return interval, $D$ varies by 0.024, on average. These results reconfirm the earlier results by indicating that the accuracy of temporal upscaling is greatest for $f_{Rn}$, $f_A$, and $f_{\lambda Eeq}$.

## 3.4 Estimating Optimal Return Interval Thresholds

Again focusing on the flux estimates when linear interpolation was used, the return interval threshold yielding errors of less than 20% in the daily ET estimates was identified (Table 3). The 20% threshold was selected because it is the nominal uncertainty commonly associated with in situ observations such as those collected via eddy covariance. While the return interval associated with the 20% threshold varied depending on land cover type and $f$, the longest return intervals are associated with $f_{\lambda Eeq}$ followed by $f_{Rn}$, and $f_A$, which yield statistically identical results, and finally $f_{K\downarrow}$ and $f_{\lambda E0}$, which also yield statistically indistinguishable results based on t-tests at the 95% confidence level. The range of values among the various $f$ was 2 days, on average. This again indicates that the accuracy of temporal upscaling is greatest for $f_{Rn}$, $f_A$, and $f_{\lambda Eeq}$.

By plotting the average threshold return interval for each land cover class against its corresponding mean latent heat flux for that class (Fig. 10), it can be seen that length of the return interval that will result in no more than 20% error decreases with the increasing moisture flux. Like $\rho$, the relationship follows an exponential decay function. In this case, however, the curve has a lower bound of five days. Based on this, the maximum return interval that can be expected to introduce less than 20% error to the flux estimates via temporal upscaling for all land cover classes is 5 days. If a threshold of 10% relative error is used, the threshold falls to only 3 days. Importantly, since the determination of the maximum return interval was made assuming there is no error in the moisture flux used to calculate $f$, they represent the best-case scenario. In practice, any error in the flux retrieval will propagate into the interpolated flux. As a result the maximum return interval would be somewhat shorter.

## 4. Conclusions

The results of this study indicate that the day-to-day persistence of $\chi$ typically used in the temporal upscaling of satellite-based ET retrievals is quite limited. The autocorrelation of daytime means of these quantities decreases to less than 0.5 within 10 day and to less than 0.25 in 7 to 24 days depending on land cover class. More generally, it was found that the number of days for $\rho$ to reach to a given threshold decreases with increasing $\lambda E$ following a well-defined exponential decay

function. This suggests that the utility of temporal upscaling is limited to short return intervals, especially for land covers such as forest and croplands, which are characterized by large moisture fluxes. The analyses of RMSE and MAE confirm this inference; in both cases the magnitude of the error increases rapidly with increasing return interval and typically reaches 75% of the maximum error within 3 to 7 days. Again, the magnitude of the error due to temporal upscaling was greatest over those land cover types with the highest ET. Using 20% relative error as the threshold, the maximum return interval ranged between five and eight days, on average, depending on land cover type. However, since the maximum return interval decreases to a minimum of five days following an exponential decay function of the mean moisture flux, five days is the longest return interval that would allow for accurate ET estimates over all land cover types assuming perfect retrieval. While the study found that using $\lambda E_{eq}$, $R_n$, or $A$ as $\chi$ tended to produce the most accurate estimates of $\lambda E$ for longer return intervals, for return intervals of five days or less, there was no statistically significant difference in the flux estimates. Finally, the comparison of interpolation methods indicated there is no advantage to using the more computationally complex spline interpolation methods.

**Acknowledgements:**

The authors would like to thank NASA for support of this research [NNH13AW37I]. They would also like to acknowledge the Ameriflux network and the investigators who contributed the data used in this study. Funding for Ameriflux is provided by the U.S. Department of Energy's Office of Science. Ameriflux data are available at http://ameriflux.ornl.gov/. The USDA is an equal opportunity employer.

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

**Figures**

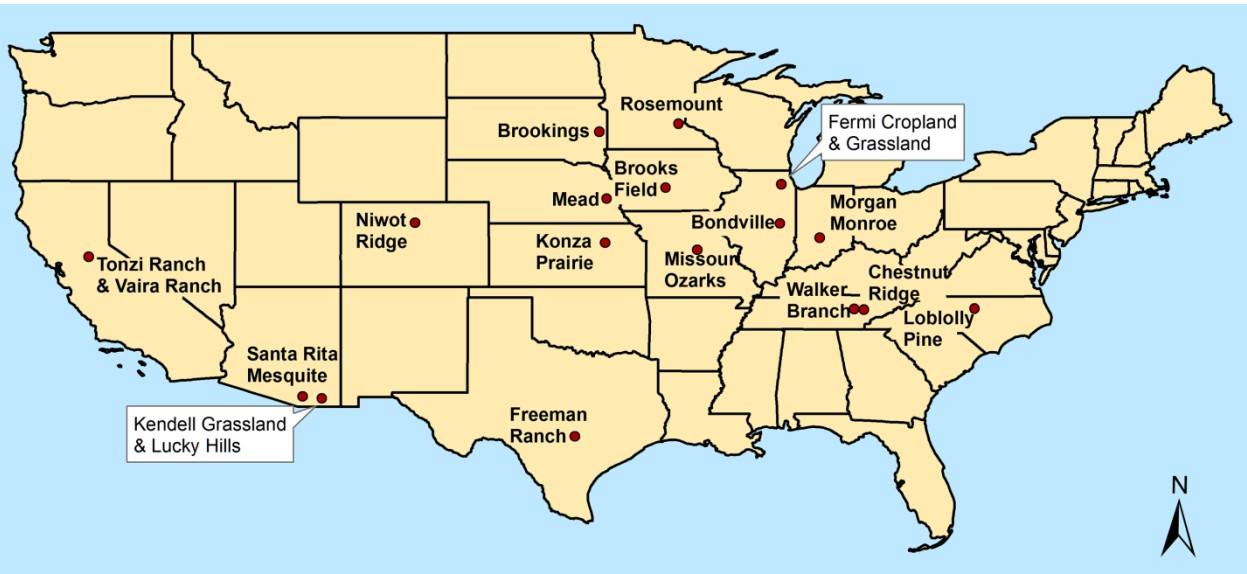

**Figure 1 Map showing the location of the Ameriflux sites used in this study.**

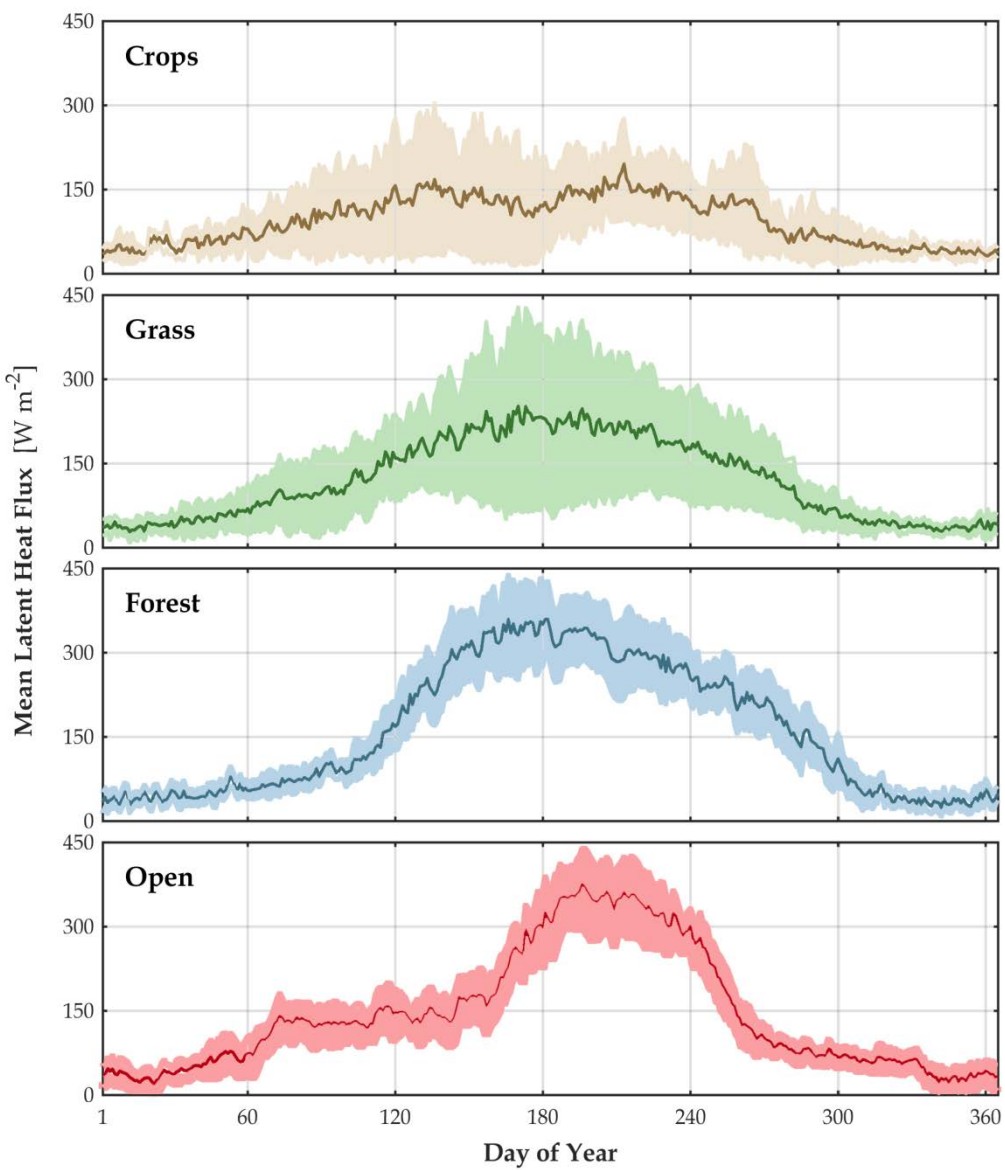

Figure 2 The mean daytime latebt heat flux is shown for each of the land cover types. The mean flux was calculated using the daytime mean flux data for all of years considered at each site. The shaded area represents one standard deviation about the mean.

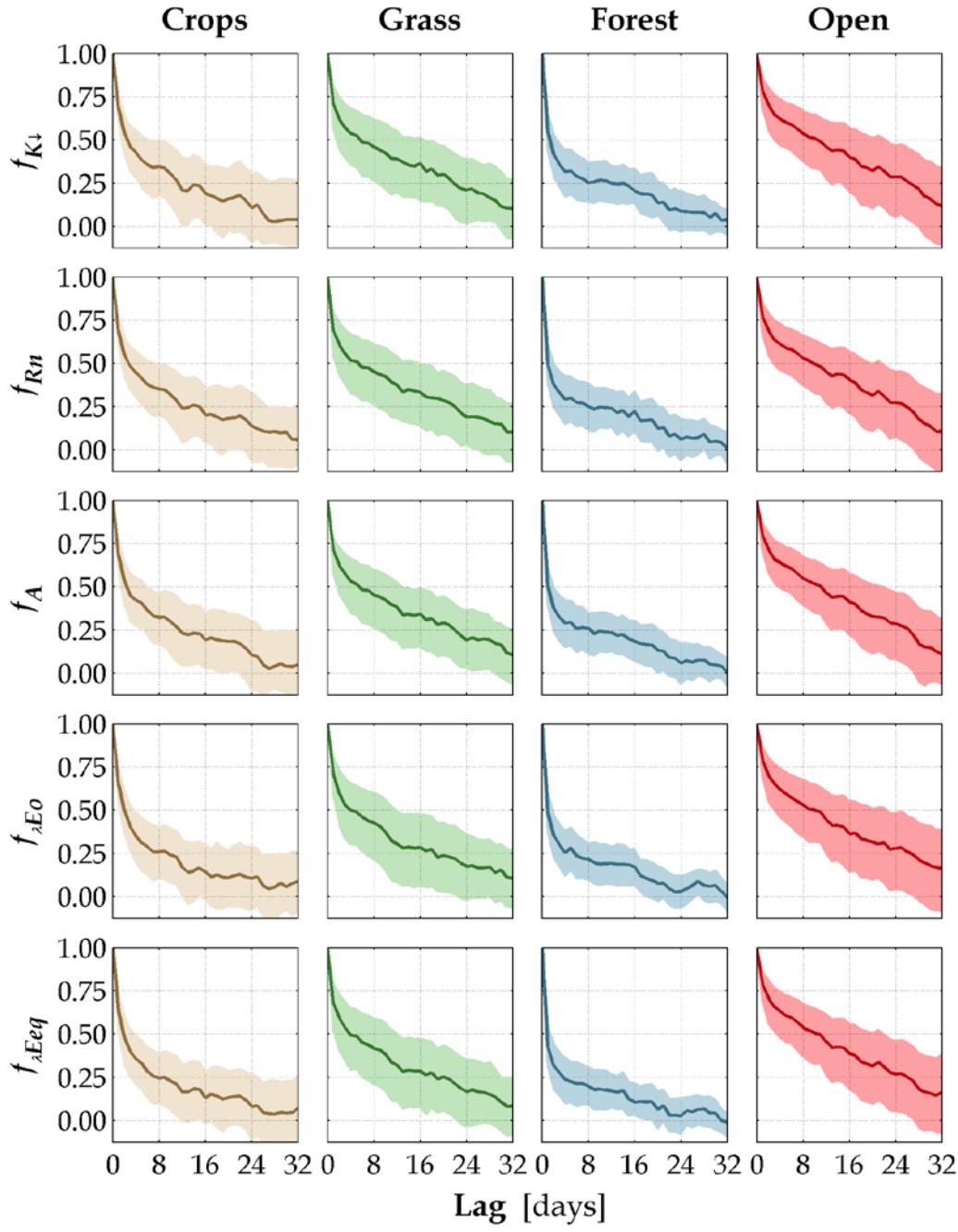

**Figure 3 The representative autocorrelation function derived for each land cover type and scaled metric used in this study is shown. The shaded area represents one standard deviation about the mean.**

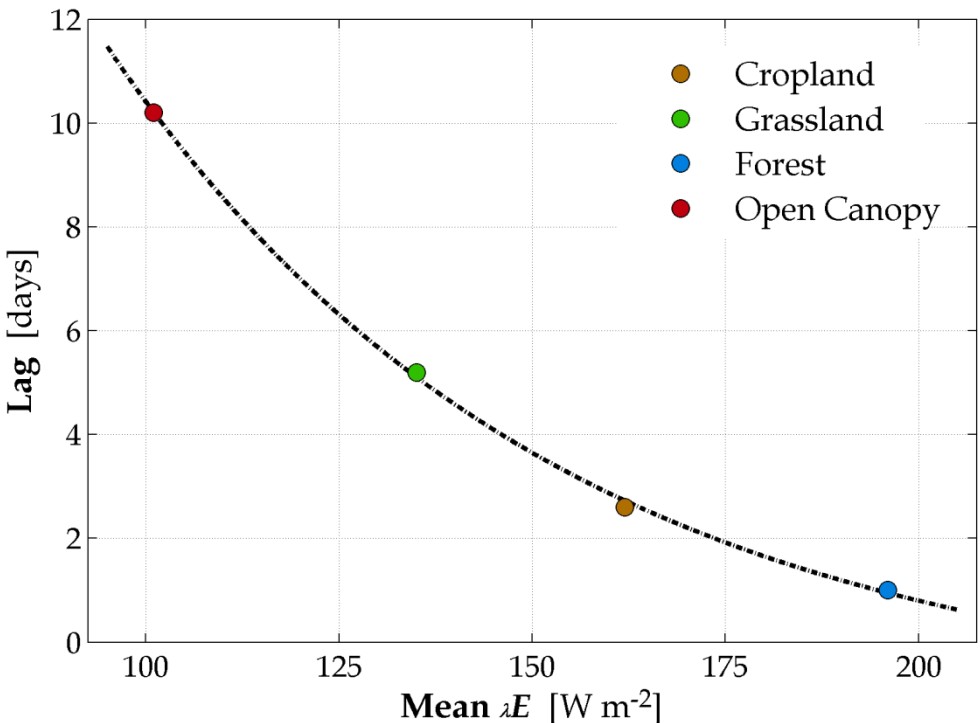

**Figure 4 The maximum lag where the autocorrelation function exceeds 0.50 plotted as a function of the mean daytime latent heat flux is shown.**

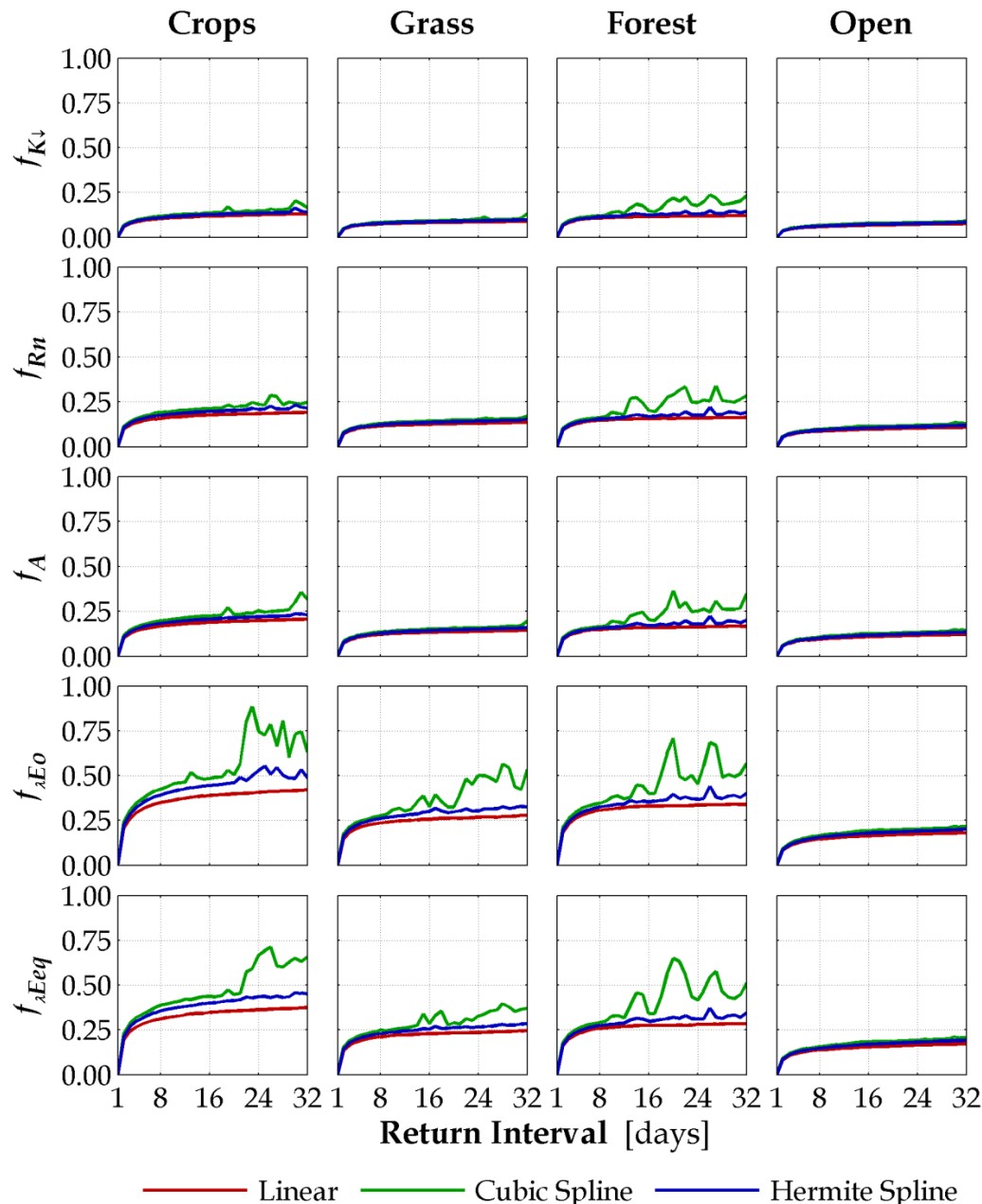

**Figure 5** The root mean square error (RMSE) of the estimates of the scaled quantities is shown for each land cover type and interpolation scheme.

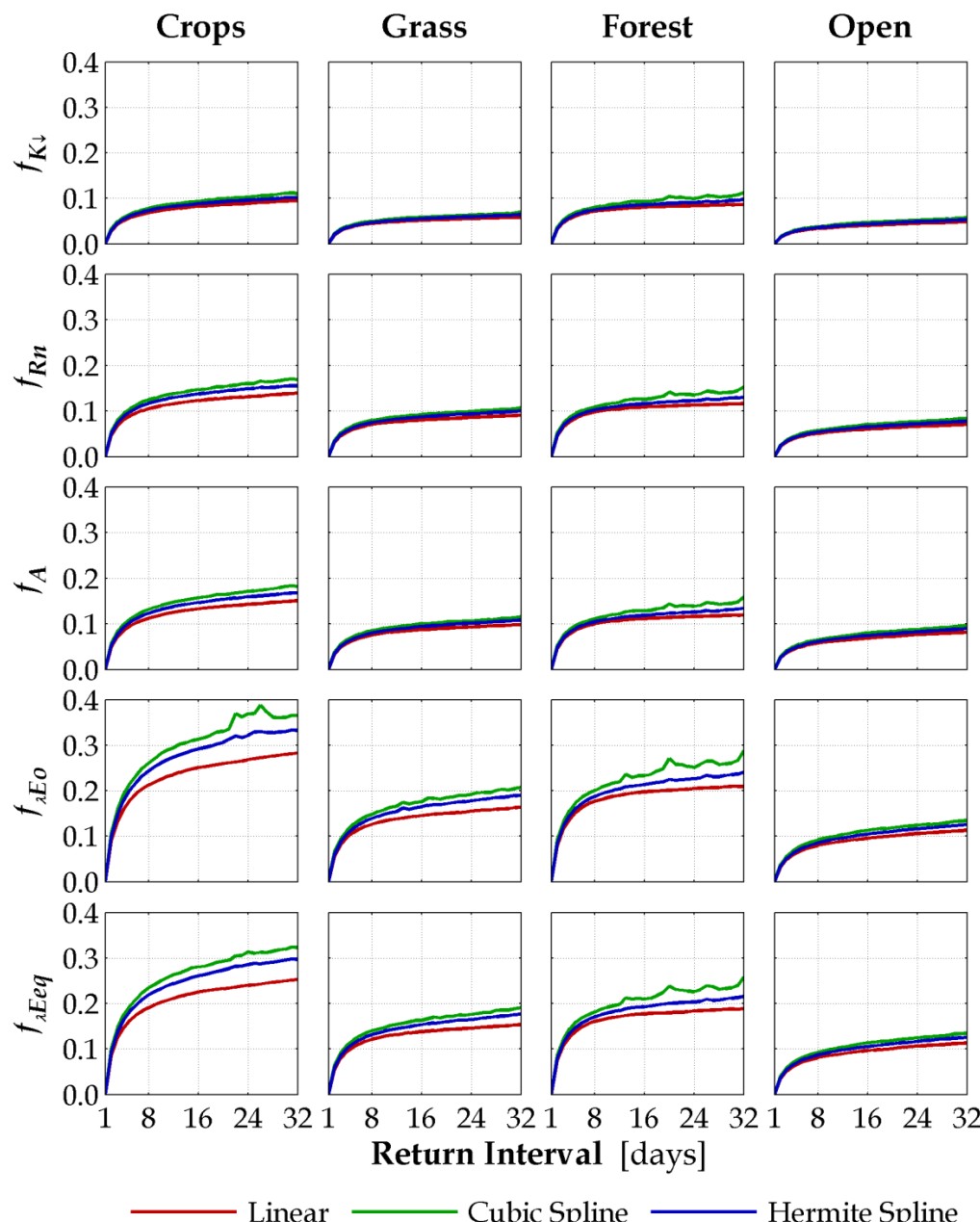

**Figure 6 The mean absolute error (MAE) of the estimates of the scaled quantities is shown for each land cover type and interpolation scheme.**

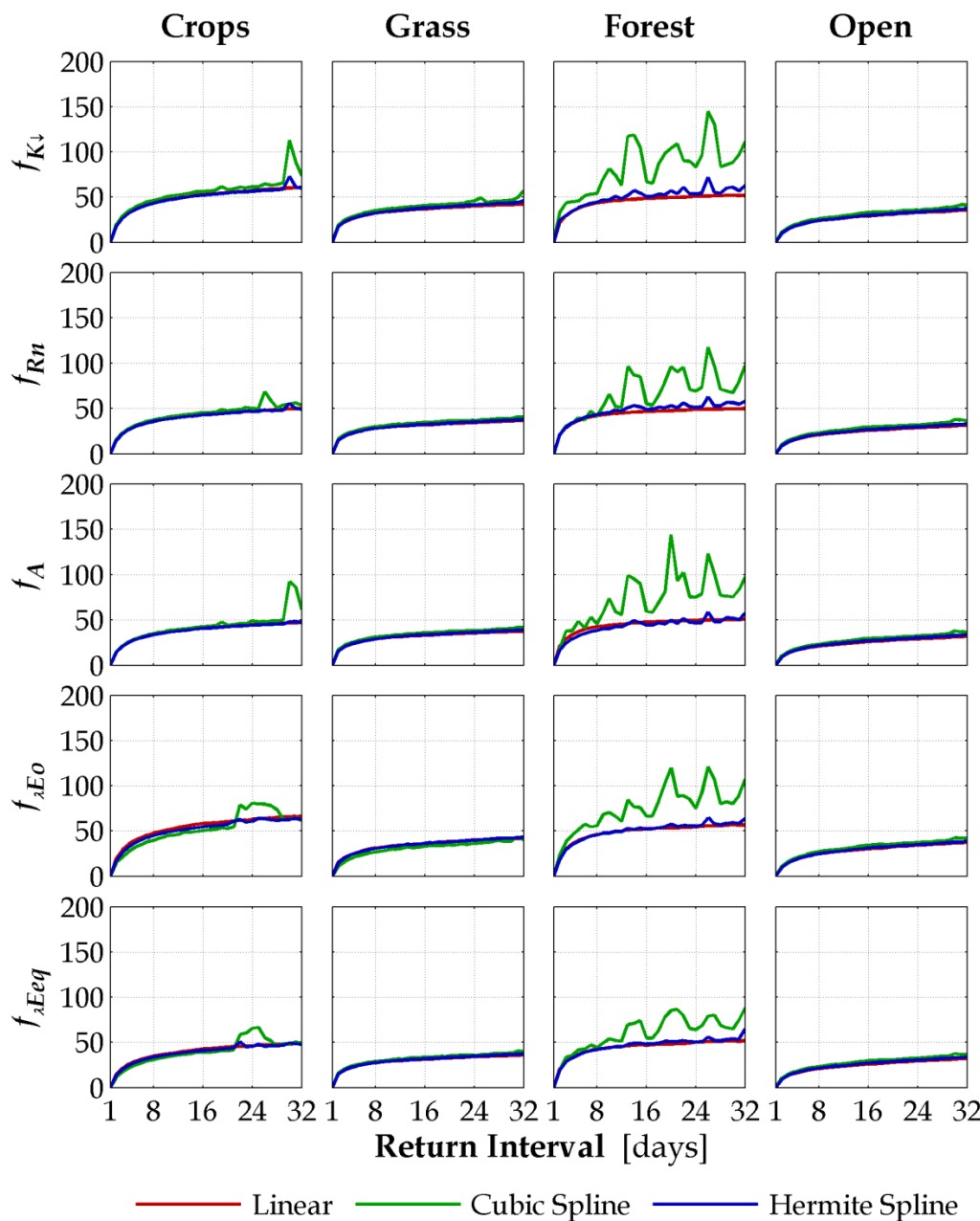

**Figure 7 The root mean square error (RMSE) of the latent heat flux derived from each of the scaled quantities is shown for each land cover type and interpolation scheme.**

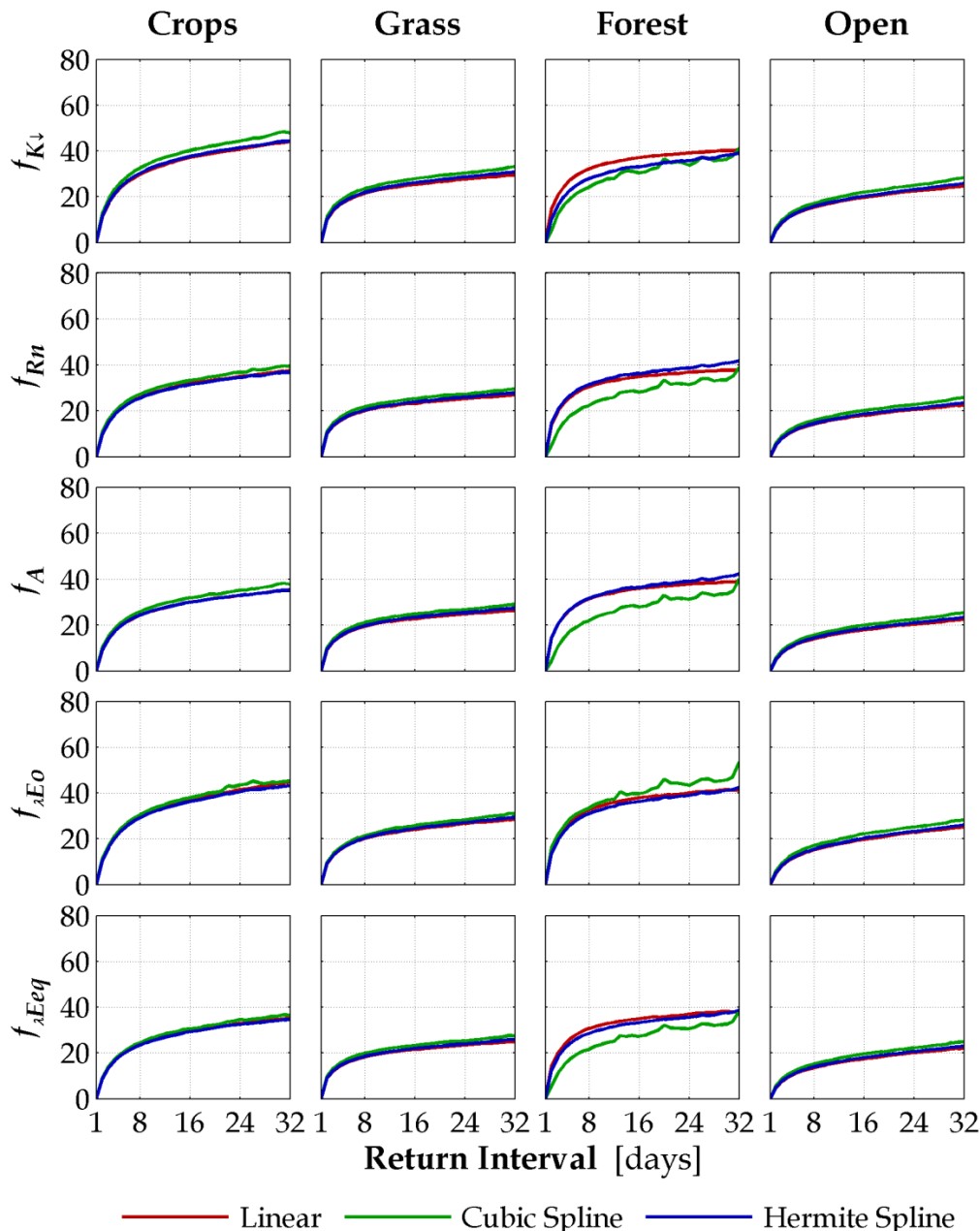

**Figure 8 The mean absolute error (MAE) of the latent heat flux derived from each of the scaled quantities is shown for each land cover type and interpolation scheme.**

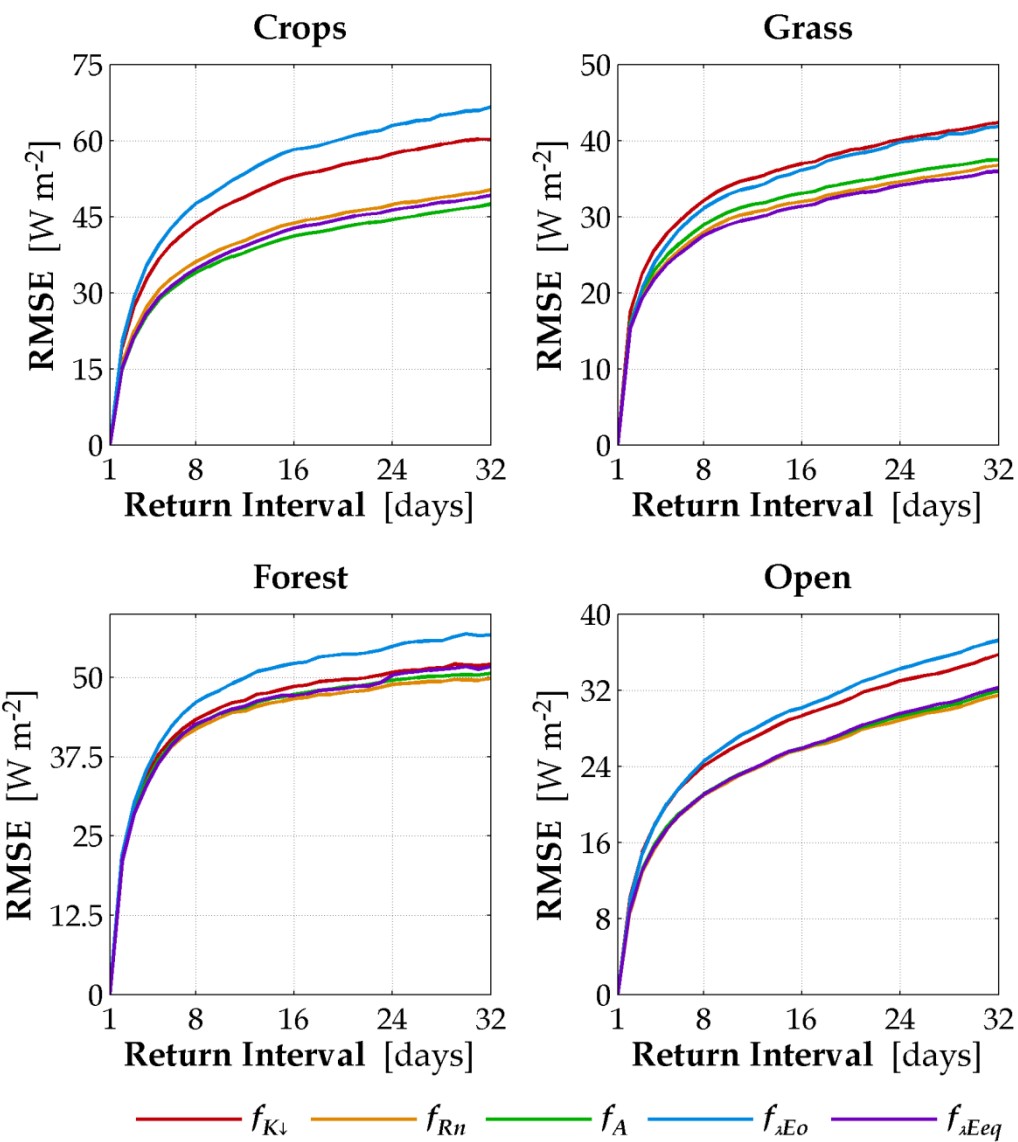

**Figure 9 The root mean square error (RMSE) of the latent heat flux derived from each of the scaled quantities is shown for each land cover type when linear interpolation is used.**

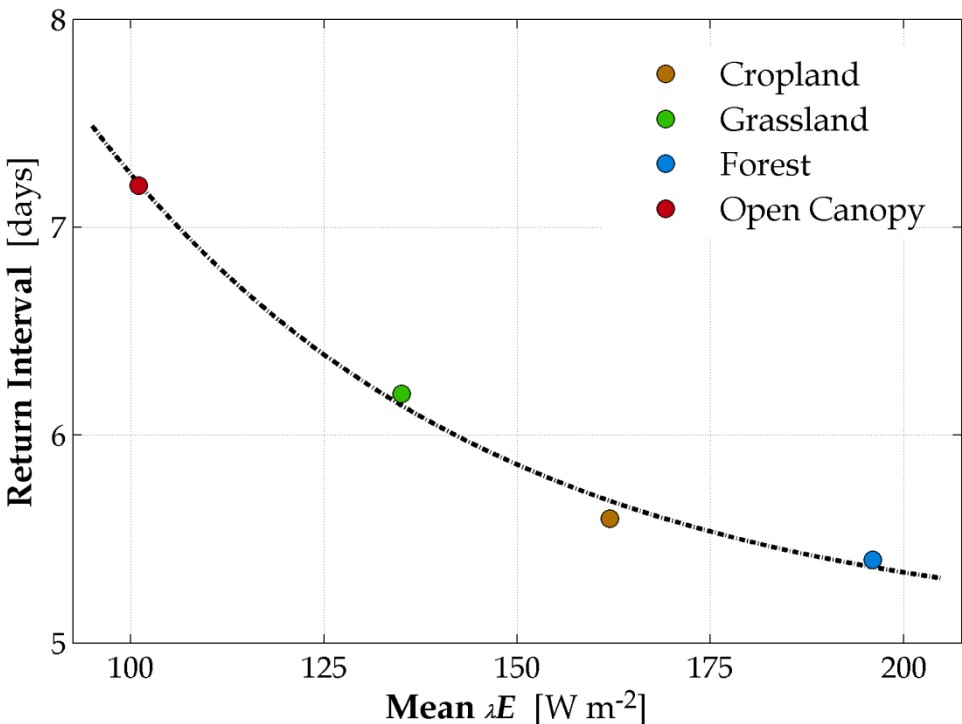

5    **Figure 10 The maximum return interval where the relative error is less than 20% plotted as a function of the mean daytime latent heat flux.**

**Tables:**

**Table 1 Summary of the 30-minute measurements collected at each Ameriflux site used in this study.**

| Meteorological Conditions | | |
|---|---|---|
| Wind Speed | Wind Direction | Air Temperature |
| Water Vapor Density | Vapor Pressure Deficit | Relative Humidity |
| Atmospheric Pressure | Precipitation | |
| **Radiation and Energy Budget** | | |
| Incident Solar Radiation | Reflected Solar Radiation | Incident Longwave Radiation |
| Terrestrial Longwave Radiation | Net Radiation | Soil Heat Flux |
| Sensible Heat Flux | Latent Heat Flux | Friction Velocity |

**Table 2 Summary of Ameriflux sites used in this study.**

| Site | Location | Land Cover | Mean Annual Temp. | Mean Annual Rainfall | Study Period | Site | Location | Land Cover | Mean Annual Temp. | Mean Annual Rainfall | Study Period |
|---|---|---|---|---|---|---|---|---|---|---|---|
| Bondville | 40.006 °N 88.290 °W | Cropland (maize/soy) | 11.0°C | 991 mm | 2000-2008 | Lucky Hills | 31.744 °N 110.052 °W | Shrubland | 17.6°C | 320 mm | 2007-2012 |
| Brookings | 44.345 °N 96.836 °W | Woody Savanna | 6.0°C | 586 mm | 2004-2010 | Mead | 41.165 °N 96.477 °W | Cropland (maize/soy) | 10.1°C | 789 mm | 2001-2012 |
| Brooks Field | 41.692 °N 93.691 °W | Cropland | 8.9°C | 847 mm | 2005-2011 | Morgan Monroe | 39.323 °N 86.413 °W | Broadleaf Deciduous Forest | 10.9°C | 1032 mm | 2004-2014 |
| Chestnut Ridge | 35.931 °N 84.332 °W | Broadleaf Deciduous Forest | 13.9°C | 1359 mm | 2005-2010 | Niwot Ridge | 40.033 °N 105.546 °W | Evergreen Needleleaf Forest | 1.5°C | 800 mm | 2001-2012 |
| Fermi Cropland | 41.859 °N 88.223 °W | Cropland (maize/soy) | 9.2°C | 929 mm | 2005-2011 | Missouri Ozarks | 38.744 °N -92.200 °W | Broadleaf Deciduous Forest | 12.1°C | 986 mm | 2004-2013 |
| Fermi Grassland | 41.841 °N 88.241 °W | Grassland | 9.2°C | 929 mm | 2005-2011 | Rosemount | 44.714 °N 93.090 °W | Cropland (maize/soy) | 6.4°C | 879 mm | 2004-2012 |
| Freeman Ranch | 29.940 °N -97.990 °W | Woody Savanna | 19.5°C | 864 mm | 2005-2009 | Santa Rita Mesquite | 31.821 °N 110.866 °W | Woody Savanna | 17.9°C | 380 mm | 2004-2012 |
| Kendall Grassland | 31.737 °N 109.942 °W | Grassland | 15.6°C | 407 mm | 2004-2012 | Tonzi Ranch | 38.432 °N 120.966 °W | Woody Savanna | 15.8°C | 559 mm | 2001-2012 |
| Konza Prairie | 39.082 °N 96.560 °W | Grassland | 12.8°C | 867 nn | 2006-2012 | Vaira Ranch | 38.407 °N 120.910 °W | Grassland | 15.8°C | 559 mm | 2001-2012 |
| Loblolly Pine | 35.978 °N 79.094 °W | Evergreen Needleleaf Forest | 14.4°C | 1170 mm | 2001-2008 | Walker Branch | 35.959 °N 84.787 °W | Broadleaf Deciduous Forest | 13.7°C | 1372 mm | 2001-2007 |

**Table 3 The maximum return interval with a relative error of less than 20% is given for each reference quantity and LULC when linear interpolation was used.**

| | | Reference Quantity | | | | |
|---|---|---|---|---|---|---|
| | | **Incident Solar Radiation** | **Net Radiation** | **Available Energy** | **Reference Latent Heat Flux** | **Equilibrium Latent Heat Flux** |
| **Land Cover** | **Cropland** | 4 | 6 | 7 | 4 | 7 |
| | **Grassland** | 5 | 7 | 6 | 5 | 8 |
| | **Forest** | 5 | 6 | 5 | 5 | 6 |
| | **Open Canopy** | 6 | 8 | 8 | 7 | 8 |