# Peer review of "Effect of the Revisit Interval and Temporal Upscaling Methods on the Accuracy of Remotely-Sensed Evapotranspiration Estimates"

_Hydrology and Earth System Sciences, 2016_

## Referee Comment (RC1) · EE Aalbers (Referee) · 6 Aug 2016

Subject of this study is the error introduced by 'temporal upscaling' of daytime evaporation as a function of the length of the interpolation interval. Temporal upscaling is used to estimate evaporation on days at which no direct evaporation data is available as E = f*X. X, the reference quantity, is known on a daily basis, the scaling factor f is assumed to vary only gradually over time and is determined through interpolation. The analysis is based on solely in situ measurements of meteorological data, turbulent fluxes and the ground heat flux, from 20 Ameriflux sites. Interpolation intervals of 1 to 32 days and five different reference quantities and associated scaling factors are considered, using three interpolation techniques and four metrics to evaluate the self-persistence of the

scaling factor and error in the scaling factor and evaporation estimate. The results are analyzed per land cover class (mean of 4-6 sites per class). The authors found that the autocorrelation of the scaling factor is limited for lag times larger than a few days; that the error in both f and E quickly increases with increasing interpolation interval during the first week, more slowly afterwards; and that both the lag time for a certain autocorrelation threshold and the interpolation interval for a certain relative error in the evaporation estimate is smallest for forests, followed by cropland, grassland and open canopy. The authors relate the differences between land cover types to the mean latent heat flux per land cover type (which is largest for forest, smallest for open canopy) and define an exponential function relating mean evaporation per land cover type with a maximum interpolation interval to achieve a relative error of x%. Using the exponential relation they define a maximum return interval of 5 days to achieve a relative error in the evaporation estimate of maximally 20%.

The question the authors address in this paper is relevant for various applications of daily evaporation time series in e.g. water resources management and hydrology. The use of the relatively large set of Ameriflux sites covering different land cover types, climates, soil types, and covering many measurement years, makes the results – in potential – also more generally applicable than other literature on the subject. The authors however focus mainly on the practical implications of the results (selection of best interpolation technique and maximum return interval for a given accuracy of the evaporation estimate), than on understanding the why of the results (e.g. what actually explains the differences they find between and within land cover types and scaling approaches). This I think is a missed chance, given the data they have and the analysis they have already performed. The approach of Farah et al. (2004) for example, showing the seasonal variability of the evaporative fraction and trying to explain it, I find very useful in this respect as an additional analysis. Moreover although 20 Ameriflux sites 'distributed throughout the contiguous United States' have been used, the only information on the differences between the sites that is provided by the authors is the land cover class. The reader therefore cannot fully judge the representativeness of the

results of this study, whereas given the practical focus of the analysis that should be a major benefit of this study. Of the comments below, at least comments 1,2,(3),7 and 8 I think should be addressed before publication of the manuscript.

**Specific comments**

1. The title of the article, mentioning the accuracy of remote sensing-based estimates of evapotranspiration, is a bit misleading. Of course the results of this study provide an indication of the effect of the revisit time on the accuracy of RS based estimates, but this study does a) not actually use remote sensing based estimates, and b) the point scale results are not one-to-one transferable to remote sensing based results. I suggest to change the title, leaving out remote sensing.

2. In the Introduction, p5, it is stated that perfect retrieval of the flux is assumed. If indeed the results found in this study are to be transferred to remotely sensed-based evaporation estimates, could the authors elaborate a bit on how realistic this assumption is, either in the introduction or in the discussion of the results? In other words, to what extend can the results based on the in situ, point scale measurements be applied for remotely sensed, usually more spatially aggregated, estimates?

*Methods*

3. I understand daytime mean evaporation instead of daily mean evaporation is used because of the better representativeness of the instantaneous scaling factor during satellite overpass. For many applications however (hydrological), daily evaporation estimates are valuable. It would be interesting if the authors could show whether the results for daily evaporation are very different. (Although also EC measurements are not that perfect in measuring nighttime fluxes (Fisher et al. 2007).)

4. I like that different 'reference quantities' are analyzed, but are these selected because these are the ones that are used most often (so for application purposes mainly), or could the authors also elaborate a bit more on the expected differences in selfpreservation of the associated scaling factors?

5. I see value in using different evaluation metrics as long as the added value is clear. I wonder whether the authors could better explain the added value of D next to the RMSE and especially MAE, either in the methods section or in the results section. As it is presented now, D is just another metric, confirming the results of MAE.

*Results*

6. All results are shown as the mean value per land cover class. Although it is a logical choice to group the measurement sites per land cover class, I expect there to be considerable differences between the sites within a class reflecting differences in e.g. climate and other environmental conditions (as visible in Fig.1 as well). However, the authors do not comment on the 'within class' variability of the autocorrelation and for the remaining results the variability is not shown at all. I suggest to at least provide some additional information on the Ameriflux sites in Table 1 (at least climate and seasonality information) and to include information on the within class variability of all results, to inform the reader on a) the robustness and b) the representativeness of the analysis.

7. In Figure 2 and Figure 9 the maximum lag time for a certain threshold of the auto-correlation and maximum return interval for a certain threshold of the relative error in evaporation estimate are plotted against the mean latent heat flux per land cover class. First, continuing on point 7, I wonder why the authors choose to plot the results for the mean latent heat flux per land cover class and not for all individual sites? Second, does there exists a similar relation between maximum lag time or return interval and latent heat flux for the data of one site? Furthermore, I would like to know whether the authors did consider other dependencies of maximum lag time (Fig.2) and maximum return interval (Fig.9) than the mean latent heat flux?

8. Page 8-9. The authors show and describe the results concerning the RMSE and MAE of the scaling factor as a function of itself (75% of its maximum reached after

xx days). Could these values be related to the absolute value of the scaling factor as well? Or in other words, how does the relative error in f develop as a function of the lag? Idem for the evaporation estimates (Page 9).

9. I think that it may not be necessary to show Figures 5 and 6. The results for the scaling factor already provide enough ground to prefer linear interpolation above spline interpolation methods, so Figure 7 summarizes the relevant information on the development of the error in evaporation estimates with increasing interpolation interval. (Although it is interesting why the MAE in the evaporation estimate for forested sites is lowest for Rn as reference quantity (compared to the other reference quantities), whereas the MAE for the associated scaling factor is largest of all scaling factors?)

10. As I understand, in the analysis as presented here, all measurement days are grouped without differentiating between e.g. seasons, phenological stages, sowing and harvest dates, cloudiness (apart from on days that are used for interpolation) and moisture availability. I'm curious whether the authors did consider doing the analysis for e.g. the growing season only, since the variability of the evaporative fraction most likely changes with the seasons (water availability, phenology (e.g. Farah et al. 2004). For applications in water resources management one might be especially interested in evaporation during the growing season and I wonder whether the maximum interval would be different.

11. It would be informative if the authors could include some figures showing the actual latent heat flux for all land cover types or preferably sites (e.g. the average annual cycle per land cover type/site), so that the reader gets some more insight into the daily and seasonal variability of the latent heat flux and evaporative fraction itself, before showing the thereof derived analysis of self-preservation of the scaling factor and interpolation errors.

**Minor and technical comments**

12. Although one can figure out what is meant, the authors may want to go through the

introduction again and clarify / restructure the text here and there. For example page 6, lines 19-26 could be rewritten a bit more clearly. I understand it like this: Based on the daily time series time series with intervals of 2 to 32 days are generated. Per interval of length n we generate the n possible unique time series starting at the first n days of the daily series, to maximize the robustness of the statistical analysis. The generated time series were subsequently filtered for clouded days.

13: I would include the variable and unit on the y-axis, at least in the top left panel of the figure (in stead of fx, everywhere).

14. Page 4, line 15: I think Lhomme and Elguero (1999) were one of the first to describe the influence of clouds on the self-preservation of the evaporative fraction? I suggest to include the reference and maybe skip the word recent.

15. For clarity, could the formulae of the five scaling factors and reference quantities be provided? Page 5, line 26: how is the available energy defined? I suppose you use A = Rn-G? Or are other (e.g. change of energy storage in the canopy) terms involved?

16: Page 6, line 28. The threshold for a cloud free day seems to tolerate quite some cloudiness? On what is the 25% based?

17. Page 7, line 1: What is the total length of the observation time series after removal of the clouded days?

18: Page 7, line 22-26. The authors could better explain the use of the 'necessary transforms'.

20: Typos:

Page 3, Line 31: presence *of*, Line 33: referred to *as*.

Page 4, Line 22: dot behind (Van Niel et al. 2012),

Line 26: For example, [*something missing?*] and Tasumi et al,

Line 30: no dot behind time

Page 5, Line 13: Delete *Namely,*

Line 18: while/by maintaining

Page 7, Line 22: Once calculated for *the* individual the sites

Page 8. Line 13: Derived *from* meteorological data

Line 14: associated *with*

Line 28. 'Interpolated estimates of each *X*', must be 'each *f* I think?

**References**

Farah, H. O., W. G. M. Bastiaanssen, and R. A. Feddes (2004). Evaluation of the temporal variability of the evaporative fraction in a tropical watershed. *International Journal of Applied Earth Observation and Geoinformation* 5.2 (2004): pp. 129-140.

Fisher, J.B., D. D. Baldocchi, L. Misson, T. E. Dawson, and A. H. Goldstein (2007). What the towers don't see at night: nocturnal sap flow in trees and shrubs at two Ameriflux sites in California. *Tree Physiology*, 27 (4) (2007): pp. 597–610. doi: 10.1093/treephys/27.4.597.

Lhomme and Elguero (1999). Examination of evaporative fraction diurnal behaviour using a soil-vegetation model coupled with a mixed-layer model. *Hydrol. Earth Syst. Sci.*, 3 (2) (1999), pp. 259–270

---

## Referee Comment (RC2) · Anonymous Referee #2 · 20 Aug 2016

- The manuscript addresses a key challenge in the estimation of evapotranspiration from satellite data, during the interval of satellite overpasses, i.e., the temporal interpolation question. This is an important topic and has many practical implication, including water management. - The main shortcoming of this paper it limits the analysis to the statistical approach only without clear discussion of the physical interpretation of (climate and surface) processes involved. E.g., how will ET varies between two satellite overpasses, and why? Such discussions may allow for more physically based upscaling methods than statistical. E.g., it is well possible to compute daily ET0 using climate data measured at ground stations, which then can be used to upscale ETa (actual) evapotranspiration derived from satellite data, e.g., see Allen et al. (2001), Mohamed

e al. (2004). - P4, l20, a bit lengthy paragraph to confirm a known fact that it is erroneous to assume clear sky condition, if it is not! - P5, l13, would be good to give map of sites to easily see different climate zones - P5, l15, give year from .... to ..... - P6, l3, Eq. 2, would have been easier to follow if the Penman-Monrtieth equation is written complete as given in the reference (Eq. 6, p24 of Allen et al., 1998), and then give the new derivation. - P6, l3, Eq. 2, what is the difference between $\lambda v$ and $\lambda$ - P6, l14, would be good to briefly describe the interpolation methods, linear, spline, and hermite. Also the description of some parameters seems very short in some places, e.g., the autocorrelation calculation . - P 7, l10, would also be good to give the % of the RMSE. - P10, l25, it would have been interesting to test the statistical results derived from the analysis against actual upscaling of satellite-based ET results - There is some spelling and grammar errors

References: - Allen, R.G., Bastiaanssen, W.G.M., Tasumi, M., Morse, A., 2001. Evapotranspiration on the watershed scale using the SEBAL model and Landsat images, ASAE Meeting Presentation, Paper Number 01-2224, Sacramento, California, USA, July 30–August 1, 2001,.

- Mohamed, Y.A., Bastiaanssen, W.G.M., Savenije, H.H.G., 2004, Spatial variability of evaporation and moisture storage in the swamps of the upper Nile studied by remote sensing techniques. J. Hydrology 289:145–164.

---

## Author Comment (AC1) · 17 Oct 2016

Effect of the Revisit Interval on the Accuracy of Remote Sensing-based Estimates of Evapotranspiration at Field Scales (hess-2016-273)

Note: The responses follow the reviewers' comments and I highlighted using red text. Excepted text from the revised manuscript is shown in blue with the changes underlined when appropriate.

**REVIEWER 1:**
Subject of this study is the error introduced by 'temporal upscaling' of daytime evaporation as a function of the length of the interpolation interval. Temporal upscaling is used to estimate evaporation on days at which no direct evaporation data is available as $E = f*X$. $X$, the reference quantity, is known on a daily basis, the scaling factor $f$ is assumed to vary only gradually over time and is determined through interpolation. The analysis is based on solely in situ measurements of meteorological data, turbulent fluxes and the ground heat flux, from 20 Ameriflux sites. Interpolation intervals of 1 to 32 days and five different reference quantities and associated scaling factors are considered, using three interpolation techniques and four metrics to evaluate the self-persistence of the scaling factor and error in the scaling factor and evaporation estimate. The results are analyzed per land cover class (mean of 4-6 sites per class). The authors found that the autocorrelation of the scaling factor is limited for lag times larger than a few days; that the error in both $f$ and $E$ quickly increases with increasing interpolation interval during the first week, more slowly afterwards; and that both the lag time for a certain autocorrelation threshold and the interpolation interval for a certain relative error in the evaporation estimate is smallest for forests, followed by cropland, grassland and open canopy. The authors relate the differences between land cover types to the mean latent heat flux per land cover type (which is largest for forest, smallest for open canopy) and define an exponential function relating mean evaporation per land cover type with a maximum interpolation interval to achieve a relative error of x%. Using the exponential relation they define a maximum return interval of 5 days to achieve a relative error in the evaporation estimate of maximally 20%.
The question the authors address in this paper is relevant for various applications of daily evaporation time series in e.g. water resources management and hydrology. The use of the relatively large set of Ameriflux sites covering different land cover types, climates, soil types, and covering many measurement years, makes the results – in potential – also more generally applicable than other literature on the subject. The authors however focus mainly on the practical implications of the results (selection of best interpolation technique and maximum return interval for a given accuracy of the evaporation estimate), than on understanding the why of the results (e.g. what actually explains the differences they find between and within land cover types and scaling approaches). This I think is a missed chance, given the data they have and the analysis they have already performed. The approach of Farah et al. (2004) for example, showing the seasonal variability of the evaporative fraction and trying to explain it, I find very useful in this respect as an additional analysis. Moreover although 20 Ameriflux sites 'distributed throughout the contiguous United States' have been used, the only information on the differences between the sites that is provided by the authors is the land cover class. The reader therefore cannot fully judge the representativeness of the results of this study, whereas given the practical focus of the analysis that should be a major benefit of this study. Of the comments below, at least comments 1,2,(3),7 and 8 I think should be addressed before publication of the manuscript.

Specific comments
1. The title of the article, mentioning the accuracy of remote sensing-based estimates of evapotranspiration, is a bit misleading. Of course the results of this study provide an indication of the effect of the revisit time on the accuracy of RS based estimates, but this study does a) not actually use remote sensing based estimates, and b) the point scale results are not one-to-one transferable to remote sensing based results. I suggest to change the title, leaving out remote sensing.

Given the significance of temporal upscaling for remote sensing-based applications, which was indeed the impetus for this study, we feel it is important to acknowledge the importance of this study to remote sensing community. However, in light of the reviewer's comments, we agree the paper's title may be confusing to some readers and have modified it to clearly indicate that surface observations were used. The revised title reads:

Insights into the Effect of the Revisit Interval on the Accuracy of Remote Sensing-based Estimates of Evapotranspiration Temporal Upscaling Studies using In Situ Observations

2. In the Introduction, p5, it is stated that perfect retrieval of the flux is assumed. If indeed the results found in this study are to be transferred to remotely sensed-based evaporation estimates, could the authors elaborate a bit on how realistic this assumption is, either in the introduction or in the discussion of the results? In other words, to what extend can the results based on the in situ, point scale measurements be applied for remotely sensed, usually more spatially aggregated, estimates?

Since any error in the modeled ET from remote sensing-based models would propagate into the estimates of the scaled quanties, *f*, used for the interpolation and then into the interpolated fluxes, the analysis conducted here represents a best-case scenario. This is now stated explicitly in the introduction. Additionally, the effects of any errors in the modeled fluxes on the maximum return interval and the conclusions drawn from the study are discussed.

Page 5, Line 11:
Since any errors in the remote sensing-based ET estimates propagate into the calculation of f and the subsequent temporal upscaling, this analysis represents the best-case scenario.

Methods
3. I understand daytime mean evaporation instead of daily mean evaporation is used because of the better representativeness of the instantaneous scaling factor during satellite overpass. For many applications however (hydrological), daily evaporation estimates are valuable. It would be interesting if the authors could show whether the results for daily evaporation are very different. (Although also EC measurements are not that perfect in measuring nighttime fluxes (Fisher et al. 2007).)

We agree that daily ET can be very useful for many applications. This decision to focus on daytime mean data is the result of a number of considerations. This research was driven by a need to characterize the effects of temporal upscaling on remote sensing-based applications. Since the internal physics of most, if not all, remote sensing-based models for estimating ET – for example, the two source energy balance models (TSEB) – to describe the moisture flux accurately at night, daytime mean data was used to emulate the model output as closely as possible. Similarly, the assumption of self-preservation that underpins temporal upscaling is not valid at night. Finally, the contribution to the total moisture flux is typically small overnight; it averages between 3% and 8% for most land cover types (Rawson and Clark 1988; Green et al. 1989; Sugita and Brutsaert 1991; Malek, 1992; Tolk et al. 2006).

4. I like that different 'reference quantities' are analyzed, but are these selected because these are the ones that are used most often (so for application purposes mainly), or could the authors also elaborate a bit more on the expected differences in self-preservation of the associated scaling factors?

The reference quantities evaluated in this study were selected because they either have been proposed for use or are commonly used by the remote sensing community for the temporal upscaling of ET derived from remotely sensed imagery. This is now stated explicitly in the manuscript.

Page 6, Line 7:
Each of the $\chi$ used is this study was selected because it is either in common usage for remote sensing-based applications or has been proposed for use in those applications.

5. I see value in using different evaluation metrics as long as the added value is clear. I wonder whether the authors could better explain the added value of D next to the RMSE and especially MAE, either in the methods section or in the results section. As it is presented now, D is just another metric, confirming the results of MAE.

As is now discussed in the paper, the index of agreement, *D*, provides similar information as RMSE and MAE in that it describes how well the interpolated data corresponds with the observations. The chief advantage of this metric is that it facilitates inter-comparisons because it quantifies the relative agreement on a scale from 0 (no agreement) to 1 (perfect agreement).

However, since it bounded between zero, which indicates no agreement, and unity, which indicates perfect agreement, $D$ both indicates the relative magnitude of the error and facilitates the comparison of the error from differing upscaling methods.

Results

6. All results are shown as the mean value per land cover class. Although it is a logical choice to group the measurement sites per land cover class, I expect there to be considerable differences between thsites within a class reflecting differences ine.g. climate and other environmental conditions (as visible in Fig.1 as well). However, the authors do not comment on the 'within class' variability of the autocorrelation and for the remaining results the variability is not shown at all. I suggest to at least provide some additional information on the Ameriflux sites in Table 1 (at least climate and seasonality information) and to include information on the within class variability of all results, to inform the reader on a) the robustness and b) the representativeness of the analysis.

Per the reviewer's suggestion, additional information describing the climate at each of the sites has been added to Table 2. Additionally, a discussion of some of the potential reasons for the observed variability with a given land cover class has been added.

        The figure also shows there was significant variability from site-to-site within a given land cover type, particularly for longer lags. Although the specific causes of these differences are not fully understood, there are number of factors that likely contribute. For example, there are difference in both species composition and climate at the various sites. Consider, as an example, the forest class which includes both coniferous and broadleaf deciduous forest. Moreover, the species composition varies even among sites of the same forest type; for example dominant species at the Niwot ridge site are Subalpine fir (Abies lasiocarpa) and Engelmann spruce (Picea engelmannii) while, as the name implies, the dominant species at the Loblolly Pine is loblolly pine (Pinus taeda). At the same time, the mean annual temperature at the forested sites ranged from 1.5 °C to 14.4 °C while the mean annual precipitation varied from 800 mm to 1372 mm. Similarly the mean annual temperature and precipitation at the cropland sites, which are all planted on a rotation of maize and soy, range between 6.4 °C and 11.0 °C and 789 mm and 991 mm, respectively.

7. In Figure 2 and Figure 9 the maximum lag time for a certain threshold of the auto- correlation and maximum return interval for a certain threshold of the relative error in evaporation estimate are plotted against the mean latent heat flux per land cover class. First, continuing on point 7, I wonder why the authors choose to plot the results for the mean latent heat flux per land cover class and not for all individual sites? Second, does there exists a similar relation between maximum lag time or return interval and latent heat flux for the data of one site? Furthermore, I would like to know whether the authors did consider other dependencies of maximum lag time (Fig.2) and maximum return interval (Fig.9) than the mean latent heat flux?

Although the same general trends are evident when the data from the individual sites are used, the relationships were rather noisy. Therefore, the means for a given land cover type was used in lieu for clarity. The aim of this paper was to explore the effects of temporal upscaling on the accuracy of ET estimates and, with that in mind, the relationships between the mean flux and both the autocorrelation or maximum return interval is shown as a guideline for selecting an appropriate return interval. They are not intended to suggest causality. Indeed, the underlying cause of the relationships is unclear.

Additionally, although it falls outside the intent of the paper, several analyses – for example, an investigation of atmospheric coupling (McNaughton and Jarvis, 1983) - were conducted in an effort to discern the fundamental cause of the relationships between the magnitude of the moisture flux and both the autocorrelation and maximum return interval. Unfortunately, these did not yield fruitful results and the data available at all of the sites is insufficient to conduct the extensive earth system modeling efforts needed to ascertain the underlying cause of the relationships.

8. Page 8-9. The authors show and describe the results concerning the RMSE and MAE of the scaling factor as a function of itself (75% of its maximum reached after xx days). Could these values be related to the absolute value of the scaling factor as well? Or in other words, how does the relative error in f develop as a function of the lag? Idem for the evaporation estimates (Page 9).

This comment is unclear. Both the text and associated figures describe the error as a function of the lag (return interval). The reference to the amount of time needed to reach 75% of the maximum is given simply to highlight how quickly the error increase for short lag times. To improve the clarity, the mean maximum RMSE and MAE is given for each land use type.

Page 10, Line 10:
For comparison, the mean maximum RMSE for each land cover type was 0.26, 0.28, and 0.17 for croplands, grasslands, forest, and open canopies, respectively. Although it also increased logarithmically, the amount of time needed for MAE to reach 75% of the peak value was more variable, ranging between 5 to 10 days. Again, for purposes of comparison, the mean maximum MAE was 0.22, 0.14, 0.16, and 0.10, respectively, for croplands, grasslands, forest, and open canopies.

9. I think that it may not be necessary to show Figures 5 and 6. The results for the scaling factor already provide enough ground to prefer linear interpolation above spline interpolation methods, so Figure 7 summarizes the relevant information on the development of the error in evaporation estimates with increasing interpolation interval. (Although it is interesting why the MAE in the evaporation estimate for forested sites is lowest for Rn as reference quantity (compared to the other reference quantities), whereas the MAE for the associated scaling factor is largest of all scaling factors?)

While we appreciate the reviewer's suggestion, we respectfully disagree. The additional figures not only reinforce the benefit of using linear interpolation, they also show how the errors associated with the scaled quantities are propagated into the final flux estimates. By comparing the figures, the reader can clearly see that the errors in the interpolated values of the scaled quantities are mirrored in the errors in flux estimates.

10. As I understand, in the analysis as presented here, all measurement days are grouped without differentiating between e.g. seasons, phenological stages, sowing and harvest dates, cloudiness (apart from on days that are used for interpolation) and moisture availability. I'm curious whether the authors did consider doing the analysis for e.g. the growing season only, since the variability of the evaporative fraction most likely changes with the seasons (water availability, phenology (e.g. Farah et al. 2004). For applications in water resources management one might be especially interested in evaporation during the growing season and I wonder whether the maximum interval would be different.

Preliminary analyses indicate there is a seasonal pattern in the magnitude of the error with the greatest discrepancy occurring during growing season when the magnitude of the fluxes is also greatest. However, when the error is considered in relative terms, the seasonal effect vanishes. As a result, the maximum return interval would not change with the time of year.

11. It would be informative if the authors could include some figures showing the actual latent heat flux for all land cover types or preferably sites (e.g. the average annual cycle per land cover type/site), so that the reader gets some more insight into the daily and seasonal variability of the latent heat flux and evaporative fraction itself, before showing the thereof derived analysis of self-preservation of the scaling factor and interpolation errors.

Per the reviewer's suggestion a figure showing the annual pattern of the mean daytime latent heat flux for each land cover type has been added to the manuscript.

[Figure]

Figure 2 The mean daytime latebt heat flux is shown for each of the land cover types. The mean flux was calculated using the daytime mean flux data for all of years considered at each site. The shaded area represents one standard deviation about the mean.

Minor and technical comments
12. Although one can figure out what is meant, the authors may want to go through the introduction again and clarify / restructure the text here and there. For example page 6, lines 19-26 could be rewritten a bit more clearly. I understand it like this: Based on the daily time series time series with intervals of 2 to 32 days are generated. Per interval of length n we generate the n possible unique time series starting at the first n days of the daily series, to maximize the robustness of the statistical analysis. The generated time series were subsequently filtered for clouded days.

Although the reviewer has it exactly right, we agree that the paragraph referred to above is awkward. It has been rewritten to improve clarity. Additionally, the entire paper has been carefully proofread and additional revisions were made when needed to improve clarity or correct typographic errors.

The text the reviewer refers to now reads:

Page 7, Line 13:
        For this analysis, temporal upscaling was conducted at each of the Ameriflux sites using all possible combinations of $f$ and interpolation methods. Specifically, it was conducted with data representing return intervals of up to 32 day generated from the daytime mean data at each site. In order to maximize the robustness of the statistical analysis, all possible realizations - the unique yet equivalent time series that can be generated from the data collected at a particular site while maintaining constant return interval – were considered in the analysis. The total number of possible realizations for a given return interval is equal to the length of the return interval. The individual realizations were generated by beginning the time series on consecutive days.

13: I would include the variable and unit on the y-axis, at least in the top left panel of the figure (in stead of fx, everywhere).

Figures 1, 2, 5, and 6 indicate the variable used by the scaling function as a subscript; for example, the scaling metric associated with the available energy is denoted $f_A$ while the one associated with net radiation is denoted $f_{Rn}$. In all cases, the scaled metric is dimensionless.

14. Page 4, line 15: I think Lhomme and Elguero (1999) were one of the first to describe the influence of clouds on the self-preservation of the evaporative fraction? I suggest to include the reference and maybe skip the word recent.

Per the reviewer's suggestion, the reference has been added. The sentence was also revised to improve clarity.

Page 4, Line 29:
Similarly, Lhomme and Elguero (1999) and later Van Niel et al. (2012) showed that the degree of self-preservation can be influenced by cloud cover.

15. For clarity, could the formulae of the five scaling factors and reference quantities be provided? Page 5, line 26: how is the available energy defined? I suppose you use A = Rn-G? Or are other (e.g. change of energy storage in the canopy) terms involved?

Per the reviewer's suggestion, available energy is now explicitly defined as the net radiation less the soil heat flux in the manuscript. The calculation of the remaining reference quantities that are derived from other measurements, i.e. $\lambda E_0$ and $\lambda E_{eq}$, are formally defined in section 2.2.

Page 4, Line 18:
For example, it is quite common to estimate ET expressed in terms of the latent heat flux ($\lambda E$) using the available energy ($A$), here defined as the net radiation less the soil heat flux, as the reference quantity and evaporative fraction ($f_A$) as the scaled metric (e.g. Crago and Brutsaert , 1996; Bastiaanssen et al., 1998; Suleiman and Crago, 2004; Colaizzi et al., 2006; Hoedjes et al. 2008; van Niel et al., 2011; Delogu et al., 2012).

16: Page 6, line 28. The threshold for a cloud free day seems to tolerate quite some cloudiness? On what is the 25% based?

Preliminary analyses comparing the relatively simple model output with the observed incident solar radiation using known clear sky data showed agreement to within $\pm10\%$. To be confident that cloud cover was present, the 25% threshold was selected. This is now stated in the paper.

Clear-sky days were identified as those where the daytime mean of the measured K↓ was within 25% of the predicted value from a simple radiation model; this threshold was selected based on a preliminary analyses comparing the model results with observations on known clear-sky days.

17. Page 7, line 1: What is the total length of the observation time series after removal of the clouded days?

A sentence has been added indicating that a minimum of 1200 days was considered the analyses at each site.

Page 7, Line 27:
Although the number of days flagged due to cloudy conditions and omitted from subsequent analyses varied depending on the site and the return interval being modelled, at least 1200 days were considered for each of the analyses at each site.

18: Page 7, line 22-26. The authors could better explain the use of the 'necessary transforms'.

The section has been revised to better explain why the transforms are needed and how they were conducted. In part, it now reads:

Page 8, Line 24:
The aggregation was accomplished by calculating the arithmetic means after conducting any necessary transform. For example, both the auto-correlation and RMSE are non-additive quantities that cannot be averaged directly; instead, they must first be transformed into an additive quantity. In the case of the former, the auto-correlation was aggregated by averaging the results for the individual analysis periods at each of the sites after applying a Fisher z-transformation (Burt and Barber, 1996). Similarly, the RMSE data was averaged after first transforming it to the mean square error.

20: Typos:
Page 3, Line 31: presence of
Corrected

Page 4, Line 8:
This infrequent acquisition of imagery is due to both lengthy return intervals and the presence of cloud cover (Ryu et al., 2012; van Niel et al., 2012; Cammalleri et al., 2013).

Line 33: referred to as

Corrected

Page 4, Line 11:
To provide temporally continuous ET estimates, the moisture flux during the period between data acquisitions is often estimated using an interpolation technique commonly referred to as temporal upscaling.

Page 4, Line 22: dot behind (Van Niel et al. 2012)

This sentence has been deleted in response to a suggestion from Reviewer 2.

Line 26: For example, [something missing?] and Tasumi et al,

The spurious word "and" has been deleted.

Page 4, Line 34:
For example, Tasumi et al. (2005) proposed using the reference ET for alfalfa (ET$_r$) as $\chi$; later, Allen et al. (2007) proposed using the standardized reference evapotranspiration (ET$_0$) as $\chi$.

Line 30: no dot behind time

The period has been deleted.

Page 5, Line 2:
As a result, $f$ derived from ET$_r$ or ET$_0$ can be treated in much the same fashion as a crop coefficient and assumed to be nearly constant changing only slowly with time (Colaizzi et al., 2006; Chavez et al., 2009).

Page 5, Line 13: Delete Namely

Deleted per the reviewer's request

Page 5, Line 21:
These are $i.$ croplands (maize (Zea mays)/soy (Glycine max) rotation); $ii.$ grasslands; $iii.$ forests (evergreen needleleaf and broadleaf deciduous); and, $iv.$ open-canopy (shrubland and woody savanna).

Line 18: while/by maintaining

Corrected

Page 5, Line 26:
After forcing closure of the energy balance while maintaining a constant Bowen ratio (Twine et al. 2000) in order to more closely match the characteristics of the output from the models, the 30-minute measurements were used to calculate the various $\chi$ and $f$.

Page 7, Line 22: Once calculated for the individual the sites

Corrected

Page 8, Line 20:
Once calculated for the individual the sites, the statistics were aggregated to represent the typical results for a given land cover type.

Page 8. Line 13: Derived from meteorological data

Corrected

Page 9, Line 12:
Nonetheless, there were statistically significant, albeit modest, differences between the auto-correlation functions associated with $f$ derived from evaporative fraction analogues and those derived from meteorological data.

Line 14: associated with

Corrected

Page 9, Line 14:
Regardless of land cover, $\rho$ associated with $f_{K\downarrow}$, $f_{Rn}$, and $f_A$, tended to be greater than $\rho$ associated with either $f_{\lambda E0}$ or $f_{\lambda Eeq}$.

Line 28. 'Interpolated estimates of each X', must be 'each f' I think?

Corrected

Page 10, Line 5:
Both RMSE and MAE of the interpolated estimates of each $f$ were calculated for all land cover types and return intervals up to 32 days.

References

Farah, H. O., W. G. M. Bastiaanssen, and R. A. Feddes (2004). Evaluation of the temporal variability of the evaporative fraction in a tropical watershed. International Journal of Applied Earth Observation and Geoinformation 5.2 (2004): pp. 129-140.

Fisher, J.B., D. D. Baldocchi, L. Misson, T. E. Dawson, and A. H. Goldstein (2007). What the towers don't see at night: nocturnal sap flow in trees and shrubs at two Ameriflux sites in California. Tree Physiology, 27 (4) (2007): pp. 597–610. doi: 10.1093/treephys/27.4.597.

Lhomme and Elguero (1999). Examination of evaporative fraction diurnal behaviour using a soil-vegetation model coupled with a mixed-layer model. Hydrol. Earth Syst. Sci., 3 (2) (1999), pp. 259–270

Green, SR, McNaughton, K, Clothier, BE. 1989: Observations of night-time water use in kiwifruit vines and apple trees. Agric. For. Meteorol. 48,251–261.

Malek, E. 1992: Night-time evapotranspiration vs. daytime and 24 h evapotranspiration. J. Hydrol. 138, 119-129.

McNaughton, KG, Jarvis, PG, 1983: Predicting effects of vegetation changes on transpiration and evaporation. In: Kozlowski, TT (Ed.), Water Deficits and Plant Growth, vol. VII. Academic Press, pp. 1–47.

Rawson, HM, Clarke, JM. 1988: Nocturnal transpiration in wheat. Aust. J. Plant Physiol. 15, 397–406.

Sugita, M, Brutsaert, W. 1991: Daily evaporation over a region from lower boundary layer profiles measured with radiosondes. Water Resour. Res. 27, 747–752.

Tolk, JA, Howell, TA, Evett, SR. 2006: Nighttime evapotranspiration from alfalfa and cotton in a semiarid climate. Agron. J. 98, 730–736.

---

## Author Comment (AC2) · 17 Oct 2016

**RESPONSE TO REVIEWER COMMENTS**
Effect of the Revisit Interval on the Accuracy of Remote Sensing-based Estimates of Evapotranspiration at Field Scales (hess-2016-273)

Note: The responses follow the reviewers' comments and I highlighted using red text. Excepted text from the revised manuscript is shown in blue with the changes underlined when appropriate.

**REVIEWER 2:**
- The manuscript addresses a key challenge in the estimation of evapotranspiration from satellite data, during the interval of satellite overpasses, i.e., the temporal interpolation question. This is an important topic and has many practical implication, including water management.

The authors would like to thank the reviewer for the kind words and suggestions.

- The main shortcoming of this paper it limits the analysis to the statistical approach only without clear discussion of the physical interpretation of (cli- mate and surface) processes involved. E.g., how will ET varies between two satellite overpasses, and why?

Although a comprehensive review of the physical processes and factors controlling ET is be the scope of this paper, the introduction has been expanded to outline the key physical processes and factors controlling evapotranspiration as a foundation for understanding how ET varies over time and why temporal upscaling is needed.

Page 3, Line 2:
   As one component of a complex network of interconnected processes, evapotranspiration (ET) is influenced by numerous factors such as available energy, soil moisture, vegetation density, and humidity (Farquhar and Sharkey 1982; van de Griend and Owe 1994; Alves and Pereira 2003; Alfieri et al. 2007). For example, the amount energy available to drive ET depends on atmospheric properties, such as humidity and aerosol content, which influence atmospheric transmissivity (Brutseart 1975; Bird and Riordan 1986). The available energy is also controlled by surface properties, such as the type and density of vegetation cover and soil moisture, which influence not only the surface albedo and emissivity (Wittich 1997; Asner etal. 1998; Myneni et al.1989; Song et al. 1999; Lobell and Asner 2002), but also impact the amount of energy conducted into the ground (Friedl and Davis 1994; Kustas et al. 2000; Abu-Hamadeh 2003; Santanell and Friedl 2003). Moreover, the magnitude of the moisture flux can vary over a range of timescales in response to changes in the environmental conditions influencing ET. One example of this, which has been pointed out by Williams et al. (1998), Scott et al. (2014), and others, is the rapid and often persistent change in ET in response to a rain event.

Such discussions may allow for more physically based upscaling methods than statistical. E.g., it is well possible to compute daily ET0 using climate data measured at ground stations, which then can be used to upscale ETa (actual) evapotranspiration derived from satellite data, e.g., see Allen et al. (2001), Mohamede al. (2004).

The authors agree Reference ET (ETo) may be a useful quantity for temporal upscaling and a close parallel to the approach the reviewer suggests here is discussed in the manuscript; Reference ET is one of the quantities evaluated in the study. In contrast to Allen et al. (2001) and Mohamede al. (2004), however, ETo was not scaled by a crop coefficient. This was done in recognition that a crop coefficient, which have been derived for only a limited number of plant species, may not be available even when the vegetation composition of the remotely-sensed scene is known.

- P4, l20, a bit lengthy paragraph to confirm a known fact that it is erroneous to assume clear sky condition, if it is not!

Per the reviewer's suggestion, the paragraph, which repeats information provided previously, has been deleted for the sake of conciseness.

- P5, l13, would be good to give map of sites to easily see different climate zones

A figure showing the location of the Ameriflux sites used in this study has been added.

[Figure]

Figure 1 Map showing the location of the Ameriflux sites used in this study.

- P5, l15, give year from .... to .....

The study period for each of the sites is now included as a part of Table 2.

- P6, l3, Eq. 2, would have been easier to follow if the Penman-Monrtieth equation is written complete as given in the reference (Eq. 6, p24 of Allen et al., 1998), and then give the new derivation.

The authors respectfully disagree. Other than a few simplifications, such as expressing the absolute temperature as $T_k$ rather than $T+273$ or the vapor pressure deficit as $D$ rather than $e_s-e_a$, Eq.2 is the same equation as presented by Allen et al. expressed in terms of energy, i.e. as a latent heat flux ($\lambda E$), instead of equivalent depth. The conversion between the two expressions of the moisture flux is trivial and well known.

- P6, l3, Eq. 2, what is the difference between λv and λ

It is not clear what the reviewer is referring to here. The letter $\lambda$ appears in conjunction with the subscript v as $\lambda v$ denoting the latent heat of vaporization and in combination with the letter E as $\lambda E$, the commonly-used term denoting the latent heat flux.

- P6, l14, would be good to briefly describe the interpolation methods, linear, spline, and hermite. Also the description of some parameters seems very short in some places, e.g., the autocorrelation calculation.

Per the reviewer's suggestion, additional information describing the algorithms and implementation of the interpolation methods has been added to the manuscript.

Page 8, Line 28:

As the name implies, the piecewise linear interpolation estimates $f$ using a family of $n - 1$ linear relationships defined such that the linearly-interpolated $f$ ($\hat{f}_L$) at time $t$ is determined according to:

$$\hat{f}_{L_i}(t) = f_i + (t_{i+1} - t_i)m_i h \quad t_i \leq t \leq t_{i+1} \tag{4}$$

where $n$ is the number of observed data points, $f_i$ is the known $f$ at time $t_i$, $m_i$ is the slope of straight line relationship for the period between $t_i$ and $t_{i+1}$ defined as $m_i = (f_{i+1} - f_i)/(t_{i+1} - t_i)$, and $h$ is the time normalized between 0 and 1 and is defined as $h = (t - t_i)/(t_{i+1} - t_i)$. The piecewise cubic spline interpolation function is family of $n$ - 1 cubic polynomials defined such that the interpolated $f$ ($\hat{f}_S$) at time $t$ is determined according to:

$$\hat{f}_{S_i}(t) = f_i + a_i[(t_{i+1} - t_i)h]^3 + b_i[(t_{i+1} - t_i)h]^2 + c_i[(t_{i+1} - t_i)h] \quad t_i \leq t \leq t_{i+1} \tag{5}$$

where the coefficients $a_i$, $b_i$, and $c_i$ are determined by simultaneously solving the series of $n - 1$ equations with the constraints that the interpolation function, as well as its first and second derivatives, must be continuous and pass exactly through the known values of $f$. Similarly, the final interpolation technique, piecewise hermite cubic spline, defines the

$$\hat{f}_{H_i}(t) = (2h^3 - 3h^2 + 1)f_i + (-2h^3 - 3h^2)f_{i+1} + \cdots$$
$$h(h^2 - 2h + 1)(t_{i+1} - t_i)s_i + h(h^2 - h)(t_{i+1} - t_i)s_{i+1} \quad t_i \leq t \leq t_{i+1} \tag{6}$$

where $s_i$ is the slope of the curve at time $t_i$ (De Boor, 1994). For this study, it is calculated according to:

$$s_i = \frac{1}{2}\left(\frac{f_{i+1} - f_i}{t_{i+1} - t_i} + \frac{f_i - f_{i-1}}{t_i - t_{i-1}}\right) \tag{7}$$

and the variables are defined as above (Moler, 2004).

- P 7, l10, would also be good to give the % of the RMSE.

It is unclear what is meant here. The text referred to by the reviewer is describes the calculation of RMSE.

-P10, l25, it would have been interesting to test the statistical results derived from the analysis against actual upscaling of satellite-based ET results - There is some spelling and grammar errors

While we agree with the reviewer in principle that such an analysis has the potential to provide many useful insights, there is no remote sensing-based data set with a sufficiently high temporal resolution available. Thus, as is discussed in the manuscript, as well as other such as Ryu et al. (2012) and Cammalleri et al. (2014), in-situ observations are a necessary proxy for studies investigating the temporal upscaling of remote sensing-based flux estimates.

References:
- Allen, R.G., Bastiaanssen, W.G.M., Tasumi, M., Morse, A., 2001. Evap- otranspiration on the watershed scale using the SEBAL model and Landsat images, ASAE Meeting Presentation, Paper Number 01-2224, Sacramento, California, USA, July 30–August 1, 2001,.
-Mohamed, Y.A., Bastiaanssen, W.G.M., Savenije, H.H.G., 2004, Spatial variability of evaporation and moisture storage in the swamps of the upper Nile studied by remote sensing techniques. J. Hydrology 289:145–164.

---

## Editor Comment (EC1) · M. Coenders-Gerrits (Editor) · 2 Nov 2016

(1) About the title: I think the new proposal for the title is too long and not covering the content. What about e.g. "Effect of revisit interval and temporal upscaling methods on the accuracy of remotely-sensed evaporation estimates"?

(5) Used metrics: If D provides similar information as RMSE and MAE, but D is only scaled between 0 and 1. Then I suggest to chose one metric you prefer instead of showing 3 similar metrics.

---

## Editor Comment (EC2) · M. Coenders-Gerrits (Editor) · 2 Nov 2016

Comment on $\lambda$ and $\lambda_v$:
There is no difference between $\lambda$ and $\lambda_v$. Both are the latent heat of vaporization. I think the confusion is caused by the fact that the authors call the 'latent heat flux' $\lambda E$ instead of $\rho \lambda E$, with $\rho$ the density of water.

Comment on equation 2:
I agree with the reviewer to change equation 2 to the way the Penman-Monteith equation is denoted in Allen et al 1998 and add the density of water to the left hand site of the equation so $\lambda = \lambda_v$.

---

## Author Comment (AC3) · 14 Nov 2016

**RESPONSE TO EDITOR'S COMMENTS**
Effect of the Revisit Interval on the Accuracy of Remote Sensing-based Estimates of Evapotranspiration at Field Scales (hess-2016-273)

Note: The responses follow the reviewers' comments and highlighted using red text. Excepted text from the revised manuscript is shown in blue with the changes underlined when appropriate.

**Editor's Comments 1:**

1) About the title: I think the new proposal for the title is too long and not covering the content. What about e.g. "Effect of revisit interval and temporal upscaling methods on the accuracy of remotely-sensed evaporation estimates"?

We agree that the proposed title change is both more accurate and better reflects the focus of the paper. The revised title now reads:

Effect of the Revisit Interval and Temporal Upscaling Methods on the Accuracy of Remotely-Sensed Evapotranspiration Estimates

(5) Used metrics: If D provides similar information as RMSE and MAE, but D is only scaled between 0 and 1. Then I suggest to chose one metric you prefer instead of showing 3 similar metrics.

The analyses using the index of agreement (D) have been removed from the paper. The manuscript now focuses on RMSE and MAE when describing error. The computational underpinnings and sensitivities of these two metrics are sufficiently distinct to provide unique insights into magnitude and variability of the error introduced into the ET estimates via temporal upscaling.

Page 8, Line 1 to 21:
Page 11, Line 7 to 13:
The discussion of D has been remove from sections 2.4 Statistical Metrics and
3.3 Accuracy of the Latent Heat Flux Estimates. Similarly, the figure showing how D varies with return interval has been removed.

**Editor's Comments 2:**

Comment on λ and $\lambda_v$:
There is no difference between λ and $\lambda_v$. Both are the latent heat of vaporization. I think the confusion is caused by the fact that the authors call the 'latent heat flux' λE instead of ρλE, with ρ the density of water.

Comment on equation 2:
I agree with the reviewer to change equation 2 to the way the Penman-Monteith equation is denoted in Allen et 1998 and add the density of water to the left hand site of the equation so $\lambda = \lambda_v$,

The discussion of the reference latent heat flux ($\lambda E_o$) has been modified to improve its clarity while maintaining $\lambda E$ as symbolic abbreviation for the latent heat flux. While the authors recognize in the context of an equation that $\lambda E$ might be misinterpreted by some as the product of two terms, i.e. $\lambda$ and $E$, the use of $\lambda E$ to represent the latent heat flux is well-established and commonplace in the literature. The authors are concerned that the usage of a nonstandard symbol would not only create an inconsistency with the symbols use for other related quantities, e.g. the equilibrium latent heat flux ($\lambda E_{eq}$), it would also be a potential source of confusion to readers.

Although it nearly identical the equation given in the FAO56 documentation, the reference ET ($ET_o$) calculated for this study was determined following the updated version of the relationship given Walter et al. (2005). This is now clarified in the paper and that relationship is shown. The conversion from ETo to $\lambda E_o$ is then described. In this manner, the potentially confusing situation where both $\lambda$ and $\lambda_v$ appear in a single equation can be avoided.

Other errors identified in the description were also correct.

Page 6, Line 1 to 15:
   The first of the χ derived from meteorological data, $\lambda E_0$, is derived from $ET_0$ which is described by Allen et al. (1998) as the hypothetical ET (or $\lambda E$) from a well-watered grass surface with an assumed height of 0.12 m and albedo of 0.23. It is calculated using a simplified form of the Penman-Monteith equation. For this study, the updated relationship given by Walter et al. (2005) was used:

$$ET_0 = \frac{0.408\Delta(R_n - G) + \gamma \frac{C_n}{(T+273)} U(e_s - e_a)}{\Delta + \gamma(1 + UC_d)} \tag{2}$$

where $\Delta$ is the slope of the saturation vapor pressure-temperature curve (kPa K$^{-1}$), $R_n$ is the net radiation (W m$^{-2}$), G is the soil heat flux (W m$^{-2}$), $\gamma$ is the psychrometric constant (kPa K$^{-1}$), $C_n$ is a constant (37 °C s$^2$ m$^{-2}$), $T$ is the air temperature (°C), $U$ is the wind speed (m s$^{-1}$), $e_s$ is the saturation water vapor pressure (kPa), $e_a$ is the actual water vapor pressure (kPa), and $C_d$ is a constant (0.24 s m$^{-1}$). This relationship is nearly identical to the one given in Allen et al. (1998); the two formulae differ only with regard to the assumed surface resistance. While the surface resistance is assumed to be 70 s m$^{-1}$ by Allen et al. (1998), it is assumed to be 50 s m$^{-1}$ in the later work. While modest, this modification yields improved results when the daytime moisture flux is calculated on an hourly basis (Walter et al. 2005). The result is converted to $\lambda E_0$ by multiplying by the product of the density of water and the latent heat of vaporization. Similarly, $\lambda E_{eq}$, which can be thought of as the energy-driven moisture flux that is independent of surface resistance, can be expressed according to:

$$\lambda E_{eq} = A\frac{\Delta}{\Delta + \gamma} \tag{3}$$

with the variables defined as above (McNaughton, 1976; Raupach, 2001).
* * *
Walter, I. A., Allen, R. G., Elliott, R. L., Itenfisu, D., Brown, P., Jensen, M. E., Mecham, B., Howell, T. A., Snyder, R., Echings, S., Spofford, T., Hattendorf, M., Martin, D. L., Cuenca, R. H., and Wright, J. L.: The ASCE Standardized Reference Evapotranspiration Equation. Technical Committee report to the Environmental and Water Resources Institute of the American Society of Civil Engineers from the Task Committee on Standardization of Reference Evapotranspiration, 173 pp., Reston, VA, 2005.

---

## Author Response (AR2)

**RESPONSE TO EDITOR'S COMMENTS TO THE AUTHORS**
Effect of the Revisit Interval on the Accuracy of Remote Sensing-based Estimates of Evapotranspiration at Field Scales (hess-2016-273)

Note: The responses follow the reviewers' comments and I highlighted using red text. Excepted text from the revised manuscript is shown in blue with the changes underlined when appropriate.
* * *
"The authors present a study on the effects of revisit interval and temporal upscaling method on remotely-sensed ET estimates. They investigate 5 upscaling methods and compare the results to field observations from the Ameriflux data set.

Both reviewers acknowledge the usefulness of the study, but are missing the physical interpretation rather than only focussing on the statistical analysis. I think the paper will benefit from an attempt to also find physical meaning/understanding. Furthermore, I agree with reviewer #1 that for Figure 2, it is more interesting to show the actual LE value instead of the mean LE even if the data is then more scattered. Because, what is the meaning of mean LE?

The purpose of Figure 2 is to give the reader a qualitative understanding of the variability in evapotranspiration over the different land cover types during course of the year. While we appreciate the suggestion, the authors feel that the plot of the daily mean latent heat flux better achieves this goal. The original plot not only shows the general temporal pattern of the flux for each land cover category, it is also indicates how variable that pattern is from year-to-year/site-to-site using the standard deviation.

In contrast, because the observations were collected at 20 sites over a span of 15 years, any figure attempting to show that large volume of data tends to be difficult to read and interpret. As can be seen in the example panel showing the flux from the grassland sites, even if the same 4-panel layout that was used with the current figure is used here so that each land cover type is plotted in a separate panel, the resulting plot is cluttered and confusing. It is nearly impossible to discern either the seasonal patterns or variability due the overlap of the curves for the individual sites along with the compression of the axes.

[Figure]

Note: the sample plot is scaled to represent a single panel of a full-page figure.

Specific comments:
-Table 1: a bit redundant table. Better to explain this in the text.
Per the editor's suggestion, the table has been deleted and the information incorporated into the text. The texts describing the data sets (P 5, L 18-23) now reads:

> Data, including local meteorological conditions (wind speed and direction, air temperature, humidity, atmospheric pressure, and precipitation), radiation budget (incident and reflected solar radiation, incident and terrestrial longwave radiation, and net radiation), surface fluxes (sensible, latent, and soil heat fluxes), and surface conditions, collected at numerous sites within the Ameriflux network (Baldocchi et al., 2001) were used for this study. Specifically, the data were collected at 20 Ameriflux sites (Fig. 1 and 2; Table 1) distributed across the contiguous United States and representing four distinct land cover types.

- P6L13-14: skip "(these are indicated....respectively)", since you are not using these abbreviations.
The parenthetical statement has been deleted.. (P 6, L22)

- P7L7: what is meant by 'first n-h' and 'final n-h'? Please explain.
For clarity, the text describing the calculation of the autocorrelation (P 8, L 2-6) has been modified to read:

> As discussed by Wilks (2006), persistence, *i.e.* the degree of self-preservation, can be assessed via auto-correlation ($\rho$). For a given lag ($h$), *i.e.* the offset between measurements pairs, the auto-correlation is defined according to:

$$\rho = \frac{\sum_{i=1}^{n-h}[(x_i - \bar{x}_-)(x_{i+h} - \bar{x}_+)]}{\sqrt{\sum_{i=1}^{n-h}(x_i - \bar{x}_-)^2 \sum_{i=1}^{n-h}(x_{i+h} - \bar{x}_+)^2}} \tag{8}$$

> where $n$ is the number of data points, $\bar{x}_-$ is the mean of the first $m$ data points and $\bar{x}_+$ is the mean of the final $m$ data points; $m$ is defined as the total number of data points less the length of the lag, *i.e.* $m = n - h$.

- P7L12: x in italic
It is now italicized. (P 8, L 10)

- P8L22: I would replace open canopy for open (also in figures)
The text (P 9, L 24) has been modified per the editor's suggestion to read:

> Further analysis shows differences in the mean autocorrelation functions exist between land cover types. Regardless of the scaled quantity considered, the mean autocorrelation function decreases most rapidly over forested sites and the most slowly over the open sites.

Figures 4 and 10 (revised numbering) have been updated as well. See example below.

[Figure]

- Fig 1: add on y-axis the autocorrelation function, since that is the correct label instead of f_{i}
Per the editor's request, the y-axis of Figure 3 (revised numbering) has been relabeled "Autocorrelation" and the labels indicating the associated scaled quantity has been moved to the right of the plots.

[Figure]

- Fig 3-6: add on y-axis RMSE or MAE, since that is the correct lable instead of f_{i}"

Again, the y-axis of Figures 5 to 8 (revised numbering) have been relabeled as appropriate and the labels indicating the associated scaled quantity has been moved to the right of the plots. See example below.

[revised manuscript text omitted]